# Expression of tumor antigens within an oncolytic virus enhances the anti-tumor T cell response

Mason J. Webb [1,2], Thanich Sangsuwannukul [2], Jacob van Vloten[2],
Laura Evgin [2,3,4], Benjamin Kendall[2], Jason Tonne[2], Jill Thompson[2],
Muriel Metko[2], Madelyn Moore[2,5], Maria P. Chiriboga Yerovi[2], Michael Olin[6],
Antonella Borgatti[7,8,9], Mark McNiven[10], Satdarshan P. S. Monga [11],
Mitesh J. Borad[12], Alan Melcher [13], Lewis R. Roberts [14] & Richard Vile [2,15,16] ✉

Although patients benefit from immune checkpoint inhibition (ICI) therapy in a broad variety of tumors, resistance may arise from immune suppressive tumor microenvironments (TME), which is particularly true of hepatocellular carcinoma (HCC). Since oncolytic viruses (OV) can generate a highly immune-infiltrated, inflammatory TME, OVs could potentially restore ICI responsiveness via recruitment, priming, and activation of anti-tumor T cells. Here we find that on the contrary, an oncolytic vesicular stomatitis virus, expressing interferon-ß (VSV-IFNß), antagonizes the effect of anti-PD-L1 therapy in a partially anti-PD-L1-responsive model of HCC. Cytometry by Time of Flight shows that VSV-IFNß expands dominant anti-viral effector CD8 T cells with concomitant relative disappearance of anti-tumor T cell populations, which are the target of anti-PD-L1. However, by expressing a range of HCC tumor antigens within VSV, combination OV and anti-PD-L1 therapeutic benefit could be restored. Our data provide a cautionary message for the use of highly immunogenic viruses as tumor-specific immune-therapeutics by showing that dominant anti-viral T cell responses can inhibit sub-dominant anti-tumor T cell responses. However, through encoding tumor antigens within the virus, oncolytic virotherapy can generate anti-tumor T cell populations upon which immune checkpoint blockade can effectively work.

Hepatocellular Carcinoma (HCC) represents the most common presentation of cancer within the liver and the 6th most common cancer worldwide[1–3]. Although immune checkpoint inhibition (ICI) therapy has produced significant survival benefits in a broad range of tumor types, a proportion of patients with putatively ICI-sensitive tumors, such as hepatocellular carcinoma, either do not respond or experience diminishing returns from ongoing therapy. The anti-PD-L1 monoclonal antibody atezolizumab in combination with the anti-vascular endothelial growth factor (VEGF) antibody bevacizumab has recently been approved as front-line therapy for HCC[4]. In the phase III imbrave150 study, this combination reduced the risk of death by 56% and the risk of disease progression by 40% compared to the treatment control arm sorafenib. However, although atezolizumab/bevacizumab has clinical activity, response rates reach about 30% and the majority of patients ultimately succumb to the disease[5].

A major factor contributing to treatment failure in HCC is believed to be the immune suppression mediated by the tumor microenvironment (TME)[6–9]. In this respect, oncolytic viruses (OV) infect and selectively replicate within tumor cells, leading to selective cancer cell lysis[10–13]. Pro-inflammatory viral oncolysis also modifies the TME and

helps to reverse immune-sparse, or -suppressive, microenvironments into highly infiltrated inflammatory immune environments[14–16]. To exploit the inflammatory aspects of OV therapy, we have developed the Vesicular Stomatitis Virus (VSV) as an oncolytic and immunother-apeutic agent against different tumor types with the rationale that delivery of VSV would disrupt the immune-suppressive TME and con-vert it into a virus-inflamed, immune infiltrated tumor site[15,17–21]. To increase both safety and immune-stimulating potency, we also expressed the IFNß gene from the virus[21–23]. Following IND-enabling toxicology studies[24,25], we conducted a first-in-human clinical study of VSV expressing human IFNß (VSV-hIFNβ) using IT injection under ultrasound guidance in patients with liver tumors [NCT01628640]. Fifteen patients were treated with a dose range of $10^5$ to $1.8 \times 10^7$ median tissue culture infectious dose ($TCID_{50}$) and the maximum tol-erated dose was $3 \times 10^6$ $TCID_{50}$. Preliminary evidence of efficacy included one partial response by Response Evaluation Criteria in Solid Tumors (RECIST) v1.1 criteria and two patients with stable disease (manuscript in preparation), making it clear that further improve-ments to the therapeutic regimen are required. Therefore, here we sought to develop the VSV-IFNß virus based on our previous studies using it as a clinical agent in order to develop its use further in com-bination with standard of care and additional immunotherapeutic strategies.

Since current standard of care HCC therapy atezolizumab/bev-acizumab incorporates ICI, in the current study we sought to develop our initial clinical approach by combining virotherapy with ICI. Therefore, here we tested the hypothesis that the heavily pro-inflammatory activity of vesicular stomatitis virus expressing interferon-ß (VSV-IFNß) virotherapy would convert the immune sup-pressive TME of HCC to an immune stimulatory environment. In turn, this would recruit/prime/activate potentially anti-tumor T cells, the activity of which anti-PD-L1 ICI therapy would be able to potentiate further, thereby generating superior anti-tumor therapy to either therapy alone[26–32]. This hypothesis is consistent with the model that inflammatory killing of tumor cells through viral oncolysis will recruit antigen-presenting cells (APC) to the tumor site, release high loads of tumor associated antigens (TAA), and facilitate the presentation of these TAA to CD8 and CD4 T cells in the draining lymph nodes, thereby generating systemically active anti-tumor T cell immunity[26–32].

To test this hypothesis, we used a slow-developing model of HCC, which resembles the etiology of human HCC[33], in which a Sleeping Beauty transposon integrates ß-catenin and hMet oncogenes into the livers of neonatal mice resulting in multi-focal tumor formation (SB-HCC) over 100-150 days. Tumor-bearing mice were partially suscep-tible to anti-PD-L1 therapy (~50% long-term cures), thereby mimicking at least one aspect of the human disease and current therapy. To our surprise, rather than observing even additive benefit to therapy, and irrespective of the relative timings of both treatments, the addition of VSV-IFNß to anti-PD-L1 therapy largely abolished the therapeutic effects of anti-PD-L1 ICI alone. We show that VSV-IFNß treatment generated a highly significant expansion of one, or a few, dominant anti-viral effector CD8+ T cell populations with the concomitant rela-tive disappearance of those putative anti-tumor T cell populations which are the target of anti-PD-L1 treatment. Based on our previous studies showing that inclusion of a tumor antigen within VSV induces CD8+ T cell-dependent anti-tumor therapy directed specifically against the virally encoded antigen[19,34–36], we hypothesized that it would be possible to amalgamate the potent anti-viral response with an anti-tumor T cell response by expressing immunologically relevant tumor-associated antigens from within the virus. Here we show that by expressing a multitude of putative, uncharacterized, tumor antigens within VSV using a cDNA library derived from three different SB-HCC explants (VSV-SB-HCC1,2,3) we observed highly significant improve-ments compared to anti-PD-L1 ICI alone - even in the absence of added anti-PD-L1 treatment.

Our data provide a cautionary message for the use of highly immunogenic viruses as tumor-specific immune-therapeutics. We show that oncolytic virotherapy can induce anti-viral T cell responses which can actively inhibit, or obscure, weak anti-tumor T cell responses leading, potentially, to decreased anti-tumor therapy. However, by chimerizing anti-viral and anti-tumor T cell responses through encoding tumor antigens within the virus, at least a proportion of the anti-viral T cell response is co-opted into an anti-tumor response. In this way, oncolytic virotherapy can be purposed for very effective immune-based tumor clearance and can generate anti-tumor T cell populations upon which immune checkpoint blockade can effectively work.

## Results

### The hMet and S45Y ß-catenin sleeping beauty transposon sys-tem induces multi-focal immune suppressive HCC

Concomitant expression of activated hMet and ß-catenin leads to multi-focal liver tumors in murine models following hydrodynamic tail vein injection[33], thereby mimicking an expression signature seen in a proportion of HCC patients[33,37,38]. We confirmed that the hMet + S45Y ß-catenin/Sleeping Beauty transposon plasmid system consistently induced multi-focal tumors in the livers of C57Bl/6 mice (Fig. 1A, B). Histologic signs of liver tumor development were observed as early as 3 weeks following hydrodynamic tail vein injection, initially with well-differentiated morphology but minimal immune infiltration. By seven weeks post injection, livers were densely populated by tumors, with large pseudo-cysts, fat deposits, and signs of extramedullary hema-topoiesis and shortly thereafter inevitably reached a terminal endpoint (Fig. 1A, B).

Whilst CD8+ T cell infiltration into livers was 10-fold higher in tumor-bearing mice than in non-tumor bearing control animals at both days 22 and 29 post hydrodynamic injection, CD4+ T cell infiltration trended towards increased levels but did not reach significance (Fig. 1C, D). The majority of liver infiltrating CD8+ and CD4+ T cells in non-tumor bearing animals had minimal expression of either Tim-3 or PD-1, although there was a small but detectable population of Tim-3+/PD-1+ CD4+ T cells (Fig. 1E, F). In contrast, the proportion of Tim-3+/PD-1+ CD8+ T cells in tumor-bearing livers at both days 22 and 29 were significantly increased (≥50% of the total CD8+ T cell population) compared to the control non-tumor bearing mice, suggesting that presence of tumor induced a profound state of CD8+ T cell dysfunc-tion/exhaustion (Fig. 1E). This change was not observed in the liver infiltrating CD4+ T cell population, where trends towards increased proportions of Tim-3+/PD-1+ CD4+ T cells did not reach significance compared to non-tumor bearing mice (Fig. 1F). Consistent with development of an immune suppressive TME within hMet + S45Y ß-catenin liver (SB-HCC) tumors, expression of PD-L1 on liver infiltrating CD11c+/MHCII+ dendritic cells was significantly increased in tumor bearing compared to non-tumor bearing mice (Fig. 1G). Finally, mac-roscopic visualization of tumor-bearing livers showed high levels of PD-L1 expression associated with the multi-focal tumor lesions which increased progressively with tumor development from 4 through 7 weeks post hydrodynamic injection (Fig. 1H). Taken together, these data indicate that SB-HCC tumors develop a highly immune-suppressive TME within 20−30 days of hydrodynamic plasmid injec-tion, thereby re-capitulating this aspect of human HCC in which a highly immunosuppressive TME[39] is associated with exhausted CD8+ T cells with reduced anti-tumor cytotoxicity[40–42].

### SB-HCC is responsive to CD8+ T cell-mediated checkpoint blockade with anti-PD-L1

Consistent with SB-HCC liver tumors developing a highly immune suppressive, PD-L1 dependent TME analogous to the human disease, treatment with anti-PD-L1 ICI led to cures of up to 50% of treated mice by day 150 post hydrodynamic injection (Fig. 2A). Therapy with

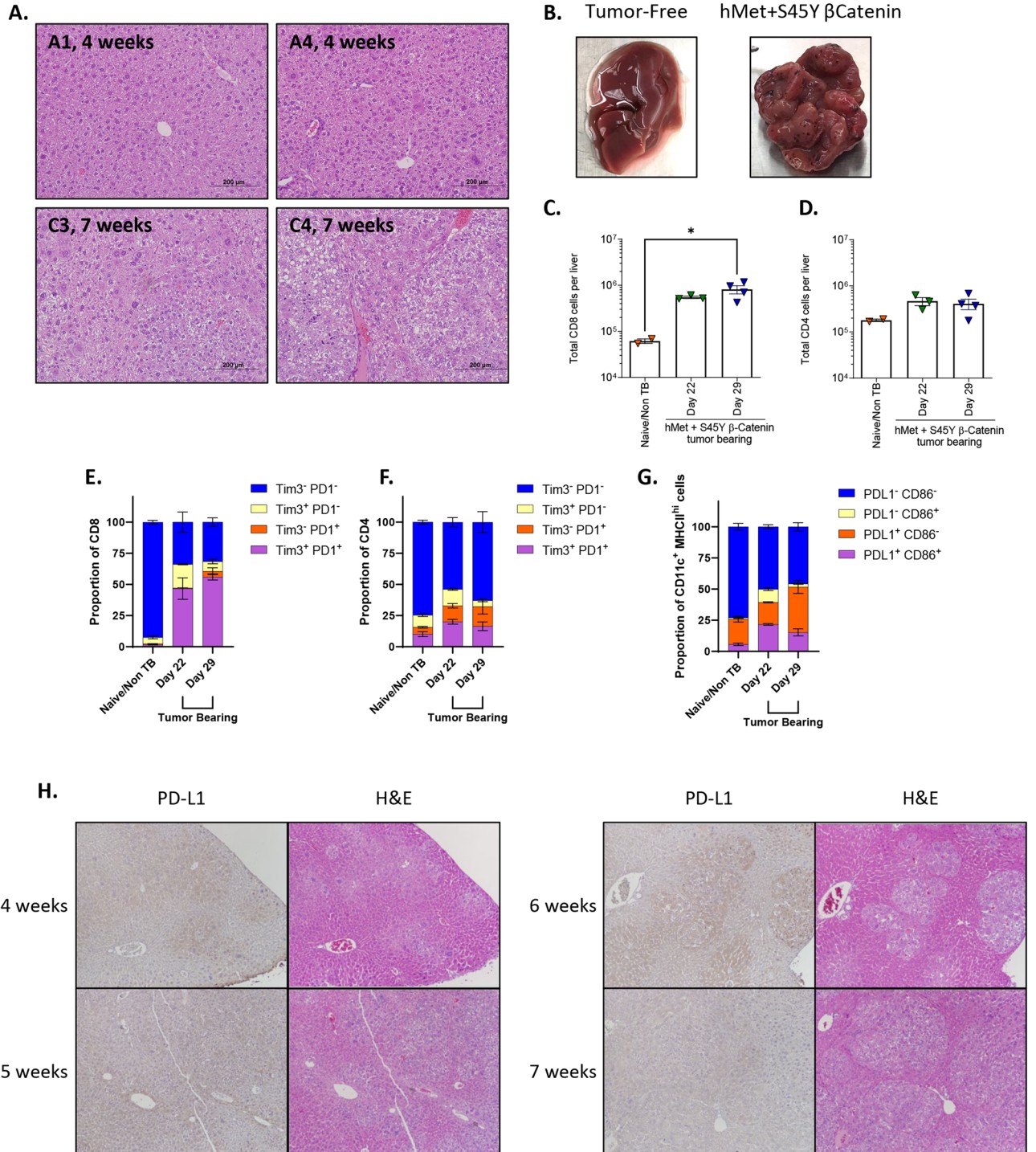

**Fig. 1 | The hMet + S45Y ß-Catenin sleeping beauty transposon system induces liver tumors which are infiltrated by PD-1⁺ CD8 T cells.** Following hydrodynamic injection of hMet + S45Y ß-Catenin plasmids, animals were euthanized at 4 and 7 weeks for pathologic studies. **A** Livers (animals A1 and A4) demonstrated appreciable neoplasia with a well-differentiated morphology and minimal immune infiltration by week 4. This progressed by week 7 (animals C1 and C4) with neo-plastic cells overtaking the majority of the tumor, with large pseudo-cysts, fat deposits, and signs of extramedullary hematopoiesis (more prominent in C4). **B** Livers from a control mouse and from a mouse 7 weeks after hMet + S45Y ß-Catenin hydrodynamic tail-vein injection. **C, D** Following hydrodynamic injection of hMet + S45Y ß-Catenin, animals were euthanized at day 22 and day 29 and analyzed for CD8+ ($p = 0.0162$) (**C**) and CD4+ ($p > 0.1$) (**D**) T cells by flow cytometry. **D** Similar to (**C**), tumor-infiltrating CD4 T cells were evaluated. **E, F** Proportions of CD8+

(Tim3- PD1- Naïve vs. Day 22 and 29, $p < 0.0001$; Tim3 + PD1+ Naïve vs. Day 22 and 29, $p < 0.0001$) (**E**) and CD4+ ($p > 0.1$ except for Tim3- PD1- Naïve vs. Day 22, $p = 0.0260$) (**F**) T cells expressing Tim3 and/or PD1⁺ in livers of tumor bearing or tumor naïve mice at days 22 and day 29. **G** Proportions of PD-L1⁺ CD86⁺ dendritic cells (DCs) in mice 22 and 29 days after hMet + S45Y ß-Catenin injection compared to tumor naïve mice. (PD-L1 + CD86+ Naïve vs. Day 22, $p = 0.0054$; PD-L1 + CD86- Naïve vs. Day 29, $p = 0.0028$; PD-L1 + CD86- Day 22 vs. Day 29, $p = 0.0002$). **H** Hematoxylin and eosin staining combined with PD-L1 immunohistochemistry of the livers of FVB mice at weeks 4, 5, 6, and 7 after hMet + S45Y ß-Catenin hydro-dynamic tail-vein injection. 10x magnification. Significance for (**C, D**) determined by ordinary one-way ANOVA, data are presented as mean values +/− SEM. Significance for (**E–G**) was determined by 2-way ANOVA with multiple comparisons using Tukey's multiple comparisons test, data are presented as mean values +/− SEM.

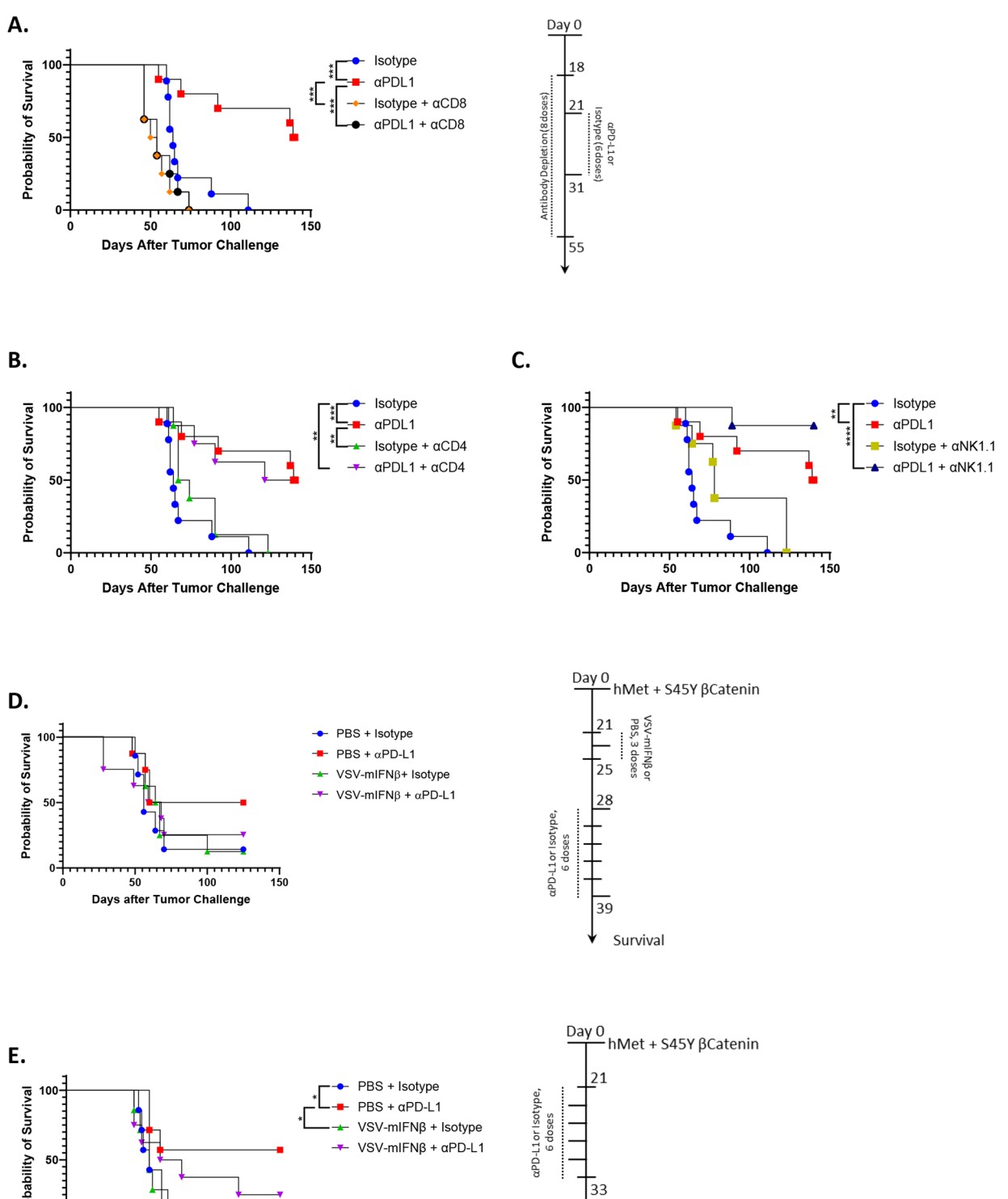

**Fig. 2 | Responsiveness of SB-HCC to CD8-mediated immune checkpoint blockade is abolished by VSV virotherapy.** Animals in all arms underwent hydrodynamic injection of hMet + S45Y ß-Catenin at day 0. $N = 67$ animals (**A**–**C**). **A**–**C** Control IgG or anti-PD-L1 ICI therapy was initiated at day 21 for 6 doses (days 21, 23, 25, 28, 30, 31) in non-depleted mice or in mice depleted of CD8 (**A**), CD4 (**B**) or NK cells (**C**) as shown (8 total doses). Survival with time is shown. **D** Mice hydrodynamically injected with hMet + S45Y ß-Catenin at day 0 were treated with PBS or VSV-mIFNß on days 21, 23, and 25 (3 doses, $10^8$ pfu/dose), followed by control IgG isotype or anti-PD-L1 on days 28, 30, 32, 35, 37, 39 (6 doses, 200 µg per dose). Survival with time is shown. **E** Mice hydrodynamically injected with hMet + S45Y ß-Catenin at day 0 were treated with control IgG isotype or anti-PD-L1 on days 21, 23, 25, 28, 30, 33 (6 doses, 200 µg per dose) and with PBS or VSV-mIFNß on days 40, 42 and 44 (3 doses, $10^8$ pfu/dose). Survival with time is shown. Significance was determined through survival curve comparison testing using a log-rank Mantel-Cox test.

anti-PD-L1 ICI was completely dependent upon CD8+ T cells (Fig. 2A) but not on CD4+ T cells (Fig. 2B). To characterize the dynamics of the anti-tumor CD8 + T cell response in more detail, Fig. S1 shows that untreated mice bearing Sleeping Beauty HCC tumors developed CD8 + T cell responses against tumor explants which increased in magnitude with time (day 0 to day 28 following tumor initiation) using ELISPOT analysis. However, between day 28 and 49 these anti-SB-HCC CD8 + T cells responses waned significantly and by day 90—at which time point most of the control mice had succumbed to the tumor—the anti-SB-HCC tumor CD8 + T cell response was low in the single surviving animal (Fig. S1A). When SB-HCC tumor-bearing mice were treated with anti-PD-L1 immune checkpoint inhibition, anti-SB-HCC CD8 + T cell responses were maintained in strength and even slightly increased through day 49 post-tumor initiation. In addition, treatment with anti-PD-L1 ICI significantly enhanced the magnitude of the CD8 + T cell response by day 90 which was associated with the prolonged survival of mice to this time point (Fig. S1B).

Over two separate experiments, depletion of NK cells showed a trend towards increased efficacy of anti-PD-L1 therapy compared to non-depleted mice, although this did not reach significance (Fig. 2C). While not definitive, this may suggest that NK-mediated immune suppression plays a role in HCC development in this model.

### Combination VSV-IFNß oncolytic virotherapy with anti-PD-L1 ICI abolishes the therapy of ICI alone

Our initial hypothesis was that early treatment of HCC with VSV-IFNß would convert the immune-suppressive TME into a pro-inflammatory environment. In turn, this would liberate HCC tumor-associated antigens (HCC$_{TAA}$) leading to the priming of anti-HCC$_{TAA}$ T cells upon which immune checkpoint therapy could work. Contrary to this hypothesis, we observed that early treatment with VSV-IFNß (day 21–25 for three doses) prior to anti-PD-L1 ICI (day 28–39 for 6 doses) completely abolished the survival benefit generated by ICI alone and was no better than either virus alone or control treatment alone (Fig. 2D). When this sequencing was reversed and virus was administered later (day 40–44) than anti-PD-L1 ICI (day 21–33), the virus still inhibited the effects of anti-PD-L1 therapy alone (Fig. 2E), although there was a (non-significant) trend to an improvement of the combination (virus + ICI) therapy compared to the virus alone. To define the mechanistic basis of the inhibition of anti-PD-L1 therapy by virus we used ELISOT assays to follow the kinetics of the anti-tumor CD8 + T cell response. As before (Fig. S1A), untreated SB-HCC-bearing mice developed a weak, but detectable, anti-SB-HCC CD8 + T cell response with time (day 0-day 28) which significantly declined by days 49 and 52 at which time mice were succumbing to disease (Fig. S2A). However, treatment with VSV-IFNß was accompanied by both a significant loss of the anti-SB-HCC CD8 + T cell response by ELISPOT reactivity to live SB-HCC explanted tumor cells and the acquisition of a potent antiviral CD8 + T cell response as measured by ELISPOT reactivity to the immunodominant VSV-N$_{52-59}$ epitope (Fig. S2B). These data suggest that treatment with oncolytic VSV-IFNß replaces a weak, slowly developing anti-tumor CD8 + T cell response with a much stronger anti-viral CD8 + T cell response.

Similarly, as before (Fig. S1B), SB-HCC tumor-bearing mice treated with anti-PD-L1 ICI developed a stronger, more prolonged anti-tumor CD8 + T cell response than in the absence of ICI (Fig. S2C). Moreover, these high levels of anti-tumor CD8 + T cell responses against SB-HCC were associated with significantly longer survival (mice surviving to day 90) (Fig. S2C). Consistent with the loss of therapy that we observed when anti-PD-L1 was combined with VSV-IFNß (Fig. 2D, E), the anti-SB-HCC CD8 + T cell response observed with both anti-PD-L1 and VSV-IFNß was significantly inhibited relative to anti-PD-L1 ICB treatment alone (Fig. S2C vs D) in mice which succumbed to disease at early time points (days 57, 61 and 90). Loss of these anti-tumor CD8 + T cell responses with combined anti-PD-L1 and VSV-IFNß treatment was

associated with the generation of potent anti-VSV CD8 + T cell responses (Fig. S2D). These data suggest that treatment with oncolytic VSV-IFNß significantly inhibits the ICI-strengthened, therapeutic, anti-tumor CD8 + T cell response by competition with a much stronger anti-viral CD8 + T cell response. Similar inhibition of the therapeutic effects of anti-PD-L1 therapy by the virus was observed with other schedules of administration including when the virus and ICI overlapped (Supplementary Fig. 3).

Taken together, these data show that the addition of a highly immunogenic oncolytic virus to a weak, anti-PD-L1-sensitive anti-tumor immune response acted consistently to inhibit the therapeutic anti-HCC$_{TAA}$ immune response.

### An immuno-dominant anti-viral rapid effector CD8+ T cell population replaces sub-dominant putative anti-HCC$_{TAA}$ T cells

We observed that when the virus was given early either before, or co-incidentally, with anti-PD-L1 ICI, the therapeutic effects of ICI therapy alone were lost (Fig. 2D, Supplementary Fig. 3), with the survival curves beginning to separate about 10 days after the last treatment with anti-PD-L1. Therefore, we used Cytometry by Time of Flight (CyTOF) analysis using t-distributed Stochastic Neighbor Embedding (tSNE) with Rphenograph to analyze 22 tumor-infiltrating lymphocyte populations (TIL) (Fig. 3A). Using differential immune marker expression, we defined a total of 9 distinct immune populations and reanalyzed through tSNE with FlowSOM, those 9 populations (Fig. 3B). When comparing TIL populations between naïve and HCC tumor-bearing mice, the presence of developing HCC tumor in the liver induced expansion of a CD8+ T cell population with markers of terminal effector cells, as defined by high Tim-3 (CD366) and PD-1 expression (Fig. 3A), at day 38 post-hydro-dynamic injection (Fig. 3C, D). This population, which we identify as 'Exhausted CD8 T Cells', correlates to cluster 17 in our 22-population tSNE analysis (Fig. 3A). This tumor-expanded population also expressed high levels of Lymphocyte Activation Gene-3 (LAG3, CD223), T-cell immunoglobulin and mucin domain-3 (TIM3), T cell immunoreceptor with Ig and ITIM domains (TIGIT), programmed cell death protein 1 (PD-1) and CD39 which are also associated with terminal effector and exhausted CD8+ T cells[43–45]. These data validate the use of SB-HCC as a model for human HCC, as the TME of HCC is enriched with exhausted, PD-1 expressing CD8+ T cells that represent the main subset of TIL and display anti-tumor cytotoxic activity in HCC[46–48]. Although the markers FoxP3, CD25/IL2r, and CD127/IL7R were included in the CyTOF panel, we were unable to identify predominant populations of Treg within the data set from these SB-HCC tumors (Fig. 3). Since the CyTOF data represents only relative abundances of different lymphocyte populations it is not possible to exclude the presence of a functionally important Treg population within these tumors and further detailed studies using other methods are underway to characterize these cell types.

Treatment of HCC-bearing mice with ICI alone (days 21–34) induced detectable changes in the landscape of CD8+ T cell populations, with a particularly significant further proportional increase in the 'Exhausted CD8 T Cells' (Cluster 17) population which is characterized by activation/inhibitory markers (TIM-3, TIGIT, LAG-3, CD39, and PD-1) (Fig. 3E). We hypothesize this population to be anti-HCC$_{TAA}$ CD8+ T cells upon which ICI therapy was operative.

In contrast, treatment of HCC-bearing mice with ICI and VSV-IFNß—under which conditions the therapeutic effects of anti-PD-L1 ICI therapy were abolished by addition of virus (Fig. 2D, E and Supplementary Fig. 3)—induced a predominant population of putative viral specific CD8+ T cells (Clusters 18, 19, 20, 22, Fig. 3A; 'Anti-viral CD8 T Cells,' Fig. 3B, F) which proportionally downregulated most of all the other CD8+ T cell populations, including the putative anti-HCC$_{TAA}$ CD8+ T cells (cluster 17, Fig. 3A; 'Exhausted CD8 T Cells,' Fig. 3B, F). This population was characterized by relatively high levels of expression of

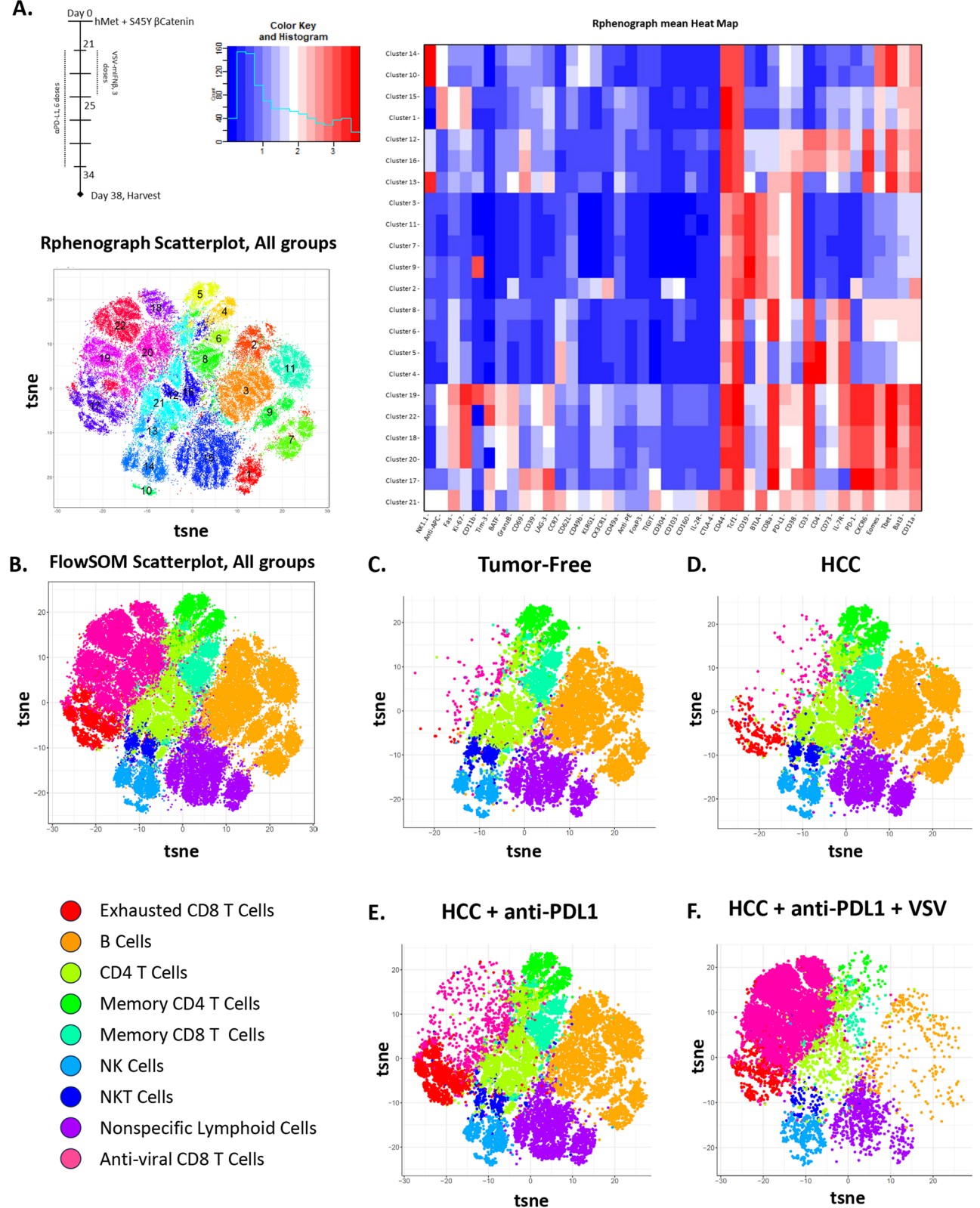

Granzyme B, interleukin 7 receptor (IL-7R, CD127), leukocyte antigen B associated transcript 3 (Bat3), and PD-L1 as well as relatively low levels of CD69 compared to the terminally differentiated putative anti-tumor CD8$^+$ T cell population−an overall profile consistent with the presence of anti-viral effector CD8$^+$ T cells. That these cells were anti-viral was confirmed using ELISPOT reactivity to the immune-dominant NSV-N$_{52-59}$ epitope (Fig. S2B, D).

Taken together, these data strongly argue that the abolition of therapeutic benefit achieved with ICI alone by the addition of virus to ICI therapy was due to a rapid induction/expansion of anti-viral effector CD8$^+$ T cells at the expense of anti-tumor, PD-1 expressing terminal effector CD8$^+$ T cells. Moreover, the repolarization of the effector response of CD8$^+$ T cells away from anti-HCC$_{TAA}$ CD8$^+$ T cells towards a population of anti-viral effector CD8$^+$ T cells would allow the

**Fig. 3 | IT treatment with VSV-mIFNß generates a dominant anti-viral CD8 T cell response which replaces the anti-tumor T cell response. A** Treatment regimen. Following hydrodynamic injection of hMet + S45Y ß-Catenin on day 0, animals were treated with anti-PD-L1 on day 21 for 6 doses concurrently with VSV-mIFNß for 3 doses. All animals were euthanized on day 38. Livers were processed into single-cell suspensions and analyzed by the Mayo CyTOF core facility. $N = 16$, 4 animals per group, groups were pooled for Rphenograph tSNE analysis, 22 groups. Scatterplot and mean heat map shown. **B** Following initial analysis (**A**), 9 distinct immune populations were identified and pooled data was re-analyzed using FlowSOM tSNE analysis. 'Exhausted CD8 T cells' were identified by expression of CD8, Tim-3, LAG-3, CD39, and PD-1 expression, 'B cells' by CD38 and CD19; 'CD4 T cells' by CD4; 'Memory CD4 T Cells' by CD4, CCR7, and CD62L; 'Memory CD8 T Cells' by CD8, CCR7, and CD62L; 'NK Cells' by NK1.1; 'NKT Cells' by NK1.1 and CD8; and 'Anti-viral CD8 T Cells' by CD8, GranzB, and PD-L1. **C** Pooled analysis of lymphocytes within livers of tumor-free mice, first 10,000 events. **D** Pooled analysis of lymphocytes within livers of SB-HCC-bearing mice, first 10,000 events. **E** Pooled analysis of lymphocytes within livers of SB-HCC bearing mice treated with anti-PD-L1, first 10,000 events. **F** Pooled analysis of lymphocytes within livers of SB-HCC-bearing mice treated with anti-PD-L1 and VSV-mIFNß, first 10,000 events.

concomitantly administered anti-PD-L1 ICI to focus upon the anti-viral, rather than anti-HCC$_{TAA}$ CD8$^+$ T cells—thereby re-invigorating the anti-viral, rather than anti-tumor, T cell response.

## Directing the anti-viral response to become an anti-tumor response

Our previous studies have shown that the inclusion of a tumor antigen within VSV induces CD8$^+$ T cell-dependent anti-tumor therapy directed specifically against the virally encoded antigen as a result of the potent immune-stimulating adjuvant properties of infection with VSV[19,34–36]. Therefore, we hypothesized that it would be possible to amalgamate the potent anti-viral response (Figs. 3 and S2) with an anti-tumor T cell response by expressing immunologically relevant tumor-associated antigens from within the virus. To test this in the hMet + S45Yß-catenin/Sleeping Beauty transposon system, we initially used the model tumor antigen OVALBUMIN (OVA) against which we could following anti-tumor T cell responses using SIINFEKL tetramer analysis. Following hydrodynamic injection of the hMet + S45Yß-catenin + pOVA/Sleeping Beauty transposon plasmid system, in which the model antigen OVA was co-expressed in the HCC tumors, anti-OVA T cells were detected in mice with hMet + S45Yß-catenin + OVA tumors from both tumor and spleen but not in naïve, non-tumor-bearing animals (Fig. 4A, B). Even though in this case OVA is a foreign non-tolerized antigen, these data were consistent with the generation of anti-HCC$_{TAA}$ T cells in these tumors as suggested from Fig. 3B–E. Furthermore, addition of anti-PD-L1 to tumor-bearing mice significantly enhanced the anti-OVA T cell response approximately threefold in both tumor and spleen, confirming that anti-PD-L1 ICI was able to expand anti-HCC$_{TAA}$ CD8$^+$ T cell responses in vivo (Fig. 4A, B).

Administration of ICI confirmed the anti-PD-L1 enhanced expansion of anti-OVA T cells in both liver and spleen (Fig. 4C–E). However, consistent with the effective replacement of anti-tumor T cells with anti-viral T cells seen in Fig. 3, the addition of either early (Fig. 3) or late (Fig. 4C, D) VSV (VSV-GFP or VSV-IFNß) to HCC-OVA tumor-bearing mice ablated almost entirely the anti-HCC$_{TAA(OVA)}$ T cell response in the liver, whether or not anti-PD-L1 ICI was also administered (Fig. 4C, D). The diminution of the anti-HCC$_{TAA(OVA)}$ T cell response by the addition of VSV-GFP or VSV-IFNß was less complete in the spleen (Fig. 4C, E) suggesting that the inflammatory anti-viral response is most dominant within the liver TME.

Therefore, we tested the hypothesis that it would be possible to amalgamate the anti-viral response with an anti-tumor response by expressing the HCC$_{TAA(OVA)}$ from within VSV. When ICI was given in the absence of virus to allow for maximal expansion of the anti-tumor T cell response, treatment with VSV-OVA alone at a late stage (Fig. 4F) generated high numbers of anti-OVA T cells in both liver and spleen (Fig. 4G, H). In addition, treatment with early anti-PD-L1 ICI combined with late VSV-OVA generated anti-HCC$_{TAA(OVA)}$ CD8$^+$ T cells to high levels as a percentage of total CD8$^+$ T cells in both liver and spleen (Fig. 4G, H). This condition of high anti-tumor CD8 + T cell levels is almost certainly a biased analysis associated with the survival of mice following ICI as opposed to that seen in mice that succumb to disease at early time points (see also Figs. S1, S2). Interestingly, the expansion of anti-HCC$_{TAA(OVA)}$ CD8$^+$ T cells by the addition of anti-PD-L1 ICI was

significantly greater than the relative expansion of anti-VSV T cells (measured by tetramer directed against the immune-dominant VSV-N$_{52-59}$ peptide of the VSV N protein) (Fig. 4G, H).

Taken together these data show that by encoding a putative HCC$_{TAA}$ within the oncolytic VSV highly significant numbers of anti-HCC$_{TAA}$ T cells were generated in vivo which were substrates for expansion by anti-PD-L1 ICI therapy and which expanded preferentially over at least one component of the immune-dominant anti-viral response.

## Selection of putative sleeping beauty TAA through prediction of high MHC class I binding epitopes

Although it was encouraging that encoding a TAA within VSV could generate ICI-expandable anti-tumor T cells, the experiments of Fig. 4 used the immunogenic model OVA antigen to which the C57Bl/6 mice were not tolerized. Therefore, to expand the concept of VSV-TAA to therapeutic efficacy against real putative HCC$_{TAA}$, RNAseq analysis of SB-HCC tumors was used to identify the top ten genes whose expression was upregulated in tumors compared to normal liver (Fig. 5A). Predicted binding affinities of peptide epitopes from these genes to the relevant H2K$^b$ and H2D$^b$ MHC Class I molecules of C57Bl/6 mice identified several strong binding epitopes from the *Lcn2*, *Lect2*, *Smagp*, *Nsdh1* and *Plrg1* genes (Fig. 5B, C), suggesting that over-expression of these genes in Sleeping Beauty HCC may expose potential neo-antigens for endogenous CD8$^+$ T cell priming/recognition. To test the ability of the three highest over-expressed genes (*Lcn2*, *Lect2*, *Smagp*) encoded separately within VSV to break tolerance to these potential HCC$_{TAA}$, VSV-IFNß co-expressing the relevant genes were tested in vivo. Splenocytes from mice treated with separate VSV-IFNß-*TAA* did not generate any detectable Th17 responses following in vitro re-stimulation with a 1:1:1 mix of three Sleeping Beauty HCC explants recovered from untreated tumor-bearing mice (SB-HCC 1,2,3) (Fig. 5D). However, splenocytes from mice vaccinated with VSV-IFNß-*Lcn2* generated a Th1-like, IFN- response to these tumor targets which was significantly greater than that generated by VSV-IFNß alone (Fig. 5E), suggesting that this TAA may be a potential HCC$_{TAA}$ in this system.

## VSV expressing a single putative HCC$_{TAA}$ may boost a pre-existing HCC$_{TAA}$ T cell response

From these data we reasoned that a treatment regimen in which early ICI was administered prior to virus treatment would optimize the expansion of the endogenous anti-tumor T cell response prior to a boosting effect with VSV-TAA. In this setting, and as before, anti-PD-L1 ICI alone cured -50% of mice bearing Sleeping Beauty HCC tumors, a therapeutic effect which was completely and rapidly eradicated by the addition of VSV-IFN-ß (Fig. 6A). Interestingly, treatment with VSV-IFNß-*Lcn2* in combination with anti-PD-L1 ICI was significantly more therapeutic than VSV-IFNß + anti-PD-L1 (Fig. 6A). However, although VSV-IFNß-*Lcn2* in combination with anti-PD-L1 did not ablate the therapy seen with anti-PD-L1 ICI alone, it did not increase therapy either (Fig. 6A). Similarly, treatment with (VSV-IFNß-*Lect2* + VSV-IFNß-*Lcn2* + VSV-IFNß-*Smagp*) in a ratio of 1:1:1 in combination with anti-PD-L1 ICI was also significantly more effective than VSV-IFNß + anti-PD-L1, but

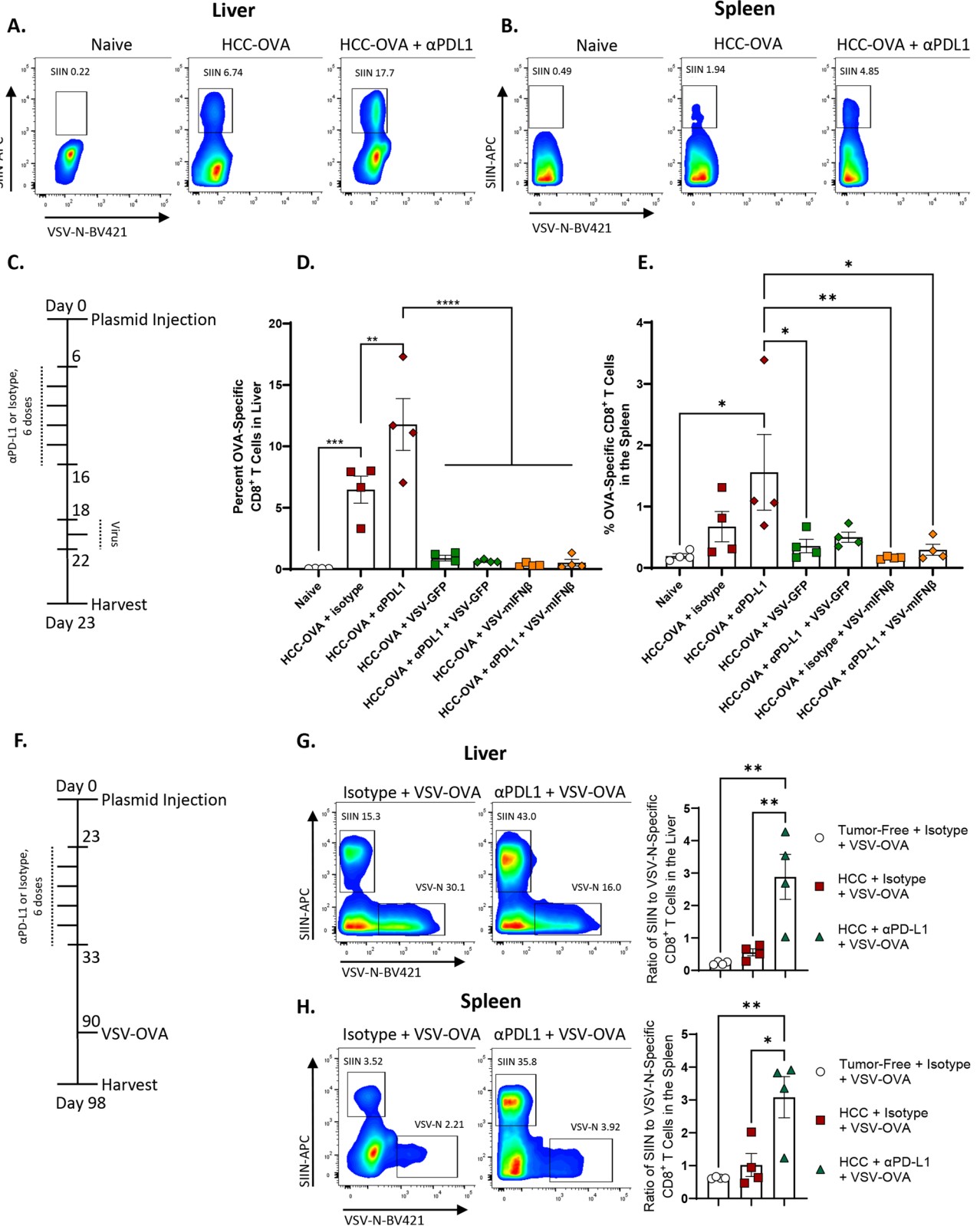

also did not improve upon, the therapy of anti-PD-L1 alone (Fig. 6A). Consistent with the results of Fig. 5D, E, splenocytes from all groups showed no detectable Th17 anti-SB-HCC 1,2,3 responses (Fig. 6B). In contrast, splenocytes from mice treated with anti-PD-L1 alone showed significant Th-1-like IFN-γ recall responses against live SB-HCC 1,2,3 explants (Fig. 6C). Splenocytes from mice treated with VSV-*Lcn2* + anti-PD-L1, showed a trend towards a recall Th-1 response against live

SB-HCC 1,2,3 explants but this did not reach significance compared to splenocytes from mice treated with VSV-IFNß + anti-PD-L1 (Fig. 6C). Finally, splenocytes from mice treated with the combination of all three putative HCC$_{TAA}$ (VSV-IFNß-*Lect2* + VSV-IFNß-*Lcn2* + VSV-IFNß-*Smagp*) + anti-PD-L1 showed no detectable response to SB-HCC 1,2,3 explants (Fig. 6C). To explain these observations, we measured the CD8 + T cell responses to each of these antigens in more detail. In

**Fig. 4 | Expression of a HCC$_{TAA}$ within VSV overcomes the loss of anti-TAA CD8 + T cells by VSV virotherapy. A, B** Following hydrodynamic injection of HCC-OVA (day 0), mice were treated with anti-PD-L1 (day 6 for 6 doses). On day 23 livers (**A**) and spleens (**B**) were processed to single-cell suspensions and analyzed by flow cytometry for SIINFEKL tetramer positive CD8 + T cells. (*n* = 12, 4 per group, representative flow data shown). **C–E** Following hydrodynamic injection of HCC-OVA (day 0), mice were treated with anti-PD-L1 (day 6 for 6 doses) followed by VSV-GFP or VSV-IFNß (3 doses, days 18,20,22). On day 23 livers (Naïve vs. HCC-OVA + isotype *p* = 0.0004, HCC-OVA + isotype vs. HCC-OVA + αPDL1 *p* = 0.0037; HCC-OVA + αPDL1 vs. HCC-OVA + VSV-GFP, HCC-OVA + αPDL1 + VSV-GFP, HCC-OVA + VSV-mIFNβ, and HCC-OVA + αPDL1 + VSV- mIFNβ *p* < 0.0001) (**D**) and spleens (Naïve vs. HCC-OVA + αPDL1 *p* = 0.0111; HCC-OVA + αPDL1 vs. HCC-OVA + VSV-GFP *p* = 0.0345, HCC-OVA + VSV-mIFNβ *p* = 0.0091, and HCC-OVA + αPDL1 + VSV-mIFNβ *p* = 0.0226) (**E**) were analyzed by flow cytometry for SIINFEKL tetramer positive CD8 + T cells. (*n* = 28, 4 animals per group, data are presented as mean values +/− SEM. **F–H** Following hydrodynamic injection of HCC-OVA (day 0), mice were treated with anti-PD-L1 (day 23, 6 doses) followed by VSV expressing oval-bumin (VSV-OVA) for 3 doses (day 90, 92, 94). On day 98 livers and spleens were analyzed by flow cytometry for SIINFEKL tetramer positive CD8 + T cells. (*n* = 12, 4 per group, data are presented as mean values +/− SEM. Representative flow data is shown as well as the pooled ratio of SIINFEKL-tetramer$^+$ to VSV-N$^+$ CD8 T cells). Tumor-Free + Isotype + VSV-OVA vs. HCC + αPDL1 + VSV-OVA *p* = 0.0032. HCC + Isotype + VSV-OVA vs. HCC + αPDL1 + VSV-OVA *p* = 0.0074. **G** Tumor-Free + Isotype + VSV-OVA vs. HCC + αPDL1 + VSV-OVA *p* = 0.0059. HCC + Isotype + VSV-OVA vs. HCC + αPDL1 + VSV-OVA *p* = 0.0162. **H** Significance was determined through ordinary one-way ANOVA with Tukey's multiple comparison tests, flow gating strategies shown in Supplementary Fig. 4.

Fig. 6C, 10 days following the last dose of VSV-*Lcn2* at dose of 3 × 10⁶ pfu (as part of the virus combination group) the anti-SB-HCC tumor response was undetectable (<20 IFNγ spots/10⁶ CD8 + T cells/48hrs) and not different from that induced by VSV-GFP (Fig. S4A). In contrast, the higher dose of 10⁷ pfu of VSV-*Lcn2* (single virus treatment) induced a significantly higher anti-tumor CD8 + T cell response (>20 IFNγ spots/ 10⁶ CD8 + T cells/48 h) at the 10 day time point (Fig. S4B). Therefore, we believe that the difference in the anti-tumor CD8 + T cell responses between the virus combination (lack of IFNγ) and single VSV-*Lcn2* virus treatment (low but detectable IFNγ) groups in Fig. 6C is a consequence of a dose response to VSV-*Lcn2* (3 × 10⁶ pfu in the combination virus treatment group compared to 10⁷ pfu in the single virus treatment group). Treatment with neither VSV-*Smagp* nor VSV-*Lect2* induced anti-tumor CD8 + T cell responses in excess of those induced by the non-specific T cell activation seen with VSV-GFP at either low or high dose of viruses (Fig. S4A, B).

## VSV expressing multiple undefined HCC$_{TAA}$ boosts anti-tumor CD8$^+$ T cell responses expanded by ICI

One possible interpretation of these data is that when at least one potentially immunogenic HCC$_{TAA}$, such as LCN2, was added to VSV-IFNß + anti-PD-L1, early treatment with anti-PD-L1 ICI allowed for expansion of potentially tumor reactive CD8$^+$ T cells (Cluster 17 in Fig. 3). Thereafter, the anti-LCN2 component of the anti-tumor T cell response could then be boosted by late vaccination with VSV-IFNß-*Lcn2*. If this model were true, we predicted that by adding multiple further HCC$_{TAA}$ to the vaccinating VSV-IFNß virus, an increased number of anti-HCC$_{TAA}$ T cells could be boosted by late VSV-IFNß-HCC$_{TAA}$ vaccination. Therefore, cDNA from three separate Sleeping Beauty explants mixed at a 1:1:1 ratio was cloned into the VSV-IFNß virus to give a viral stock of VSV-IFNß-SB-HCC 1,2,3 cDNA (Fig. 7A). Presence of the three most highly expressed SB-HCC genes, *Lcn2*, *Lect2*, and *Smagp* (identified by RNAseq, Fig. 5) in the VSV-IFNß-SB-HCC 1,2,3 cDNA library, but not in the ASMEL VSV-cDNA library constructed previously from melanoma cells[36], was confirmed by PCR. Conversely, the mela-noma associated genes gp100 or TYRP1 were abundant in the ASMEL library but were essentially undetectable in the VSV-IFNß-SB-HCC 1,2,3 cDNA library (Fig. 7B).

Treatment of SB-HCC tumor-bearing mice with a reduced dosage of anti-PD-L1 ICI (100 µg/mouse/injection instead of 200 µg/mouse/per injection) in order to accentuate any combinatorial therapeutic effects, generated some long-term survivors (Fig. 7C). In these experiments, as well as in those of Fig. 6A, we observed a bi-phasic therapeutic response to anti-PD-L1 ICI in mice in which some anti-PD-L1 treated mice succumbed to disease early (up to day 50–60 post tumor initiation) at essentially the same rate as control-treated mice, whilst a separate group of mice with ICI treatment were long term survivors (Fig. 6A, 7C). ELISPOT analysis showed that those ICI-treated mice which succumbed early to tumor all had low (<100 IFNγ spots per 10⁶ CD8 + T cells) anti-SB-HCC tumor CD8 + T cell responses, whilst long-

term survivors had significantly higher (>150 IFNγ spots per 10⁶ CD8 + T cells) anti-SB-HCC tumor CD8 + T cell responses (Fig. S5A). A similar analysis of the survival of individual untreated mice showed a trend in which the length of survival before succumbing to the tumor was associated with the magnitude of the anti-SB-HCC CD8 + T cell response (Fig. S5B). Taken together, these data show that there exists a positive correlation between whether mice succumb to disease or are cured (>d150) and detectable levels of CD8+ anti-tumor T cell reactivity in mice either treated with anti-PD-L1 ICI or left untreated (Fig. S5C).

As before, the addition of VSV expressing at least one putative HCC$_{TAA(Lcn2)}$ no longer ablated, but neither did it improve, the therapy with ICI alone (Fig. 7C). However, with Early anti-PD-L1 ICI + Late VSV-SB-HCC1,2,3, a combination of VSV and anti-PD-L1 ICI generated sig-nificantly improved therapy over anti-PD-L1 ICI treatment alone with 100% of mice surviving to day 150 (Fig. 7C). Interestingly, treatment with VSV-SB-HCC1,2,3 alone (no anti-PD-L1) was also very effective at generating long term cures (Fig. 7C). These data suggest that VSV-mediated display of multiple antigens was sufficient of itself to boost slow developing anti-HCC$_{TAA}$ CD8$^+$ T cell responses even in the absence of expansion by anti-PD-L1 treatment. Consistent with this, splenocytes from mice treated with anti-PD-L1 ICI alone re-stimulated in vitro with live SB-HCC Explants 1,2,3 cells secreted IFNγ but not IL-17 (Fig. 7D, E). In contrast, when splenocytes from mice treated with either VSV-SB-HCC1,2,3 alone (no anti-PD-L1 ICI), or with VSV-SB-HCC1,2,3 + anti-PD-L1 were re-stimulated in vitro with SB-HCC1,2,3 cells, IFNγ and IL-17 were identified as the only candidates from a screen of 26 different cytokines which were induced above that secreted by splenocytes from mice treated with ICI alone, indicating the induction of a Th17 recall response (Fig. 7D). Splenocytes from mice treated with either VSV-SB-HCC1,2,3 alone (no anti-PD-L1 ICI), or with VSV-SB-HCC1,2,3 + anti-PD-L1, generated a Th17 recall response when re-stimulated in vitro with SB-HCC cells (Fig. 7D). Splenocytes from mice treated with VSV-SB-HCC1,2,3 alone generated a Th1/IFNγ recall response that was equivalent to that from splenocytes from mice treated with anti-PD-L1 alone (Fig. 7E), whereas splenocytes from mice treated with VSV-IFNß-SB-HCC1,2,3 + anti-PD-L1 generated a sig-nificantly increased Th1/IFNγ recall response to SB-HCC cell explants (Fig. 7E). We also observed an apparent discrepancy between the lack of an anti-SB-HCC IFNγ response in mice treated with (VSV-IFNß-*Lect2* + VSV-IFNß-*Lcn2* + VSV-IFNß-*Smagp*) in Fig. 6C and the detectable anti-SB-HCC IFNγ response in Fig. 7E. This can be explained, we believe, because at day 30 after the last injection of virus the anti-tumor CD8 + T cell response induced by VSV-*Lcn2* alone, at both low (3 × 10⁶ pfu) and high (10⁷ pfu) doses, was significantly higher than at 10 days post the last injection of virus (Fig. S4A, B). Therefore, the slow developing anti-*Lcn2* CD8 + T cells response (Fig. S4A, B) was not detectable early (10 days) post-virus treatment (Fig. 6C) but was clearly detectable late (>30 days when mice succumbed to tumor in Fig. 7E) after the last virus injection. To confirm the mechanism associated

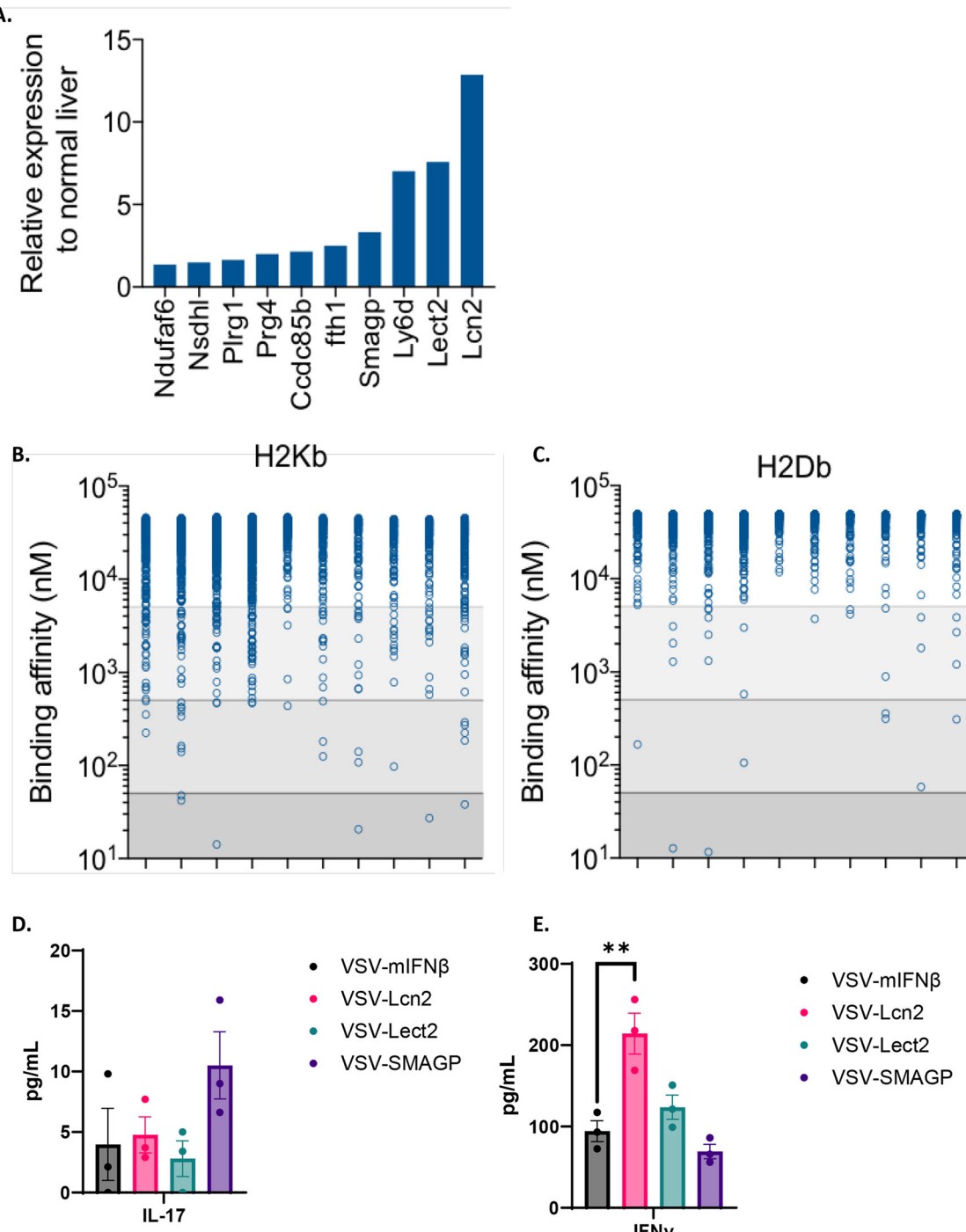

**Fig. 5 | Putative HCC_TAA selected for high-level expression and MHC binding affinity. A** SB-HCC tumor-bearing mice were euthanized on day 18 with livers harvested for RNA extraction. RNA samples were then subjected to RNA-seq analysis, identifying 10 highest relative gene expression compared to non-tumor bearing livers. **B, C** The 10 most-expressed genes identified and full-length sequences were filtered through NET MHC 2.0 binding affinity algorithm to identify octamer or nonamer peptides whose binding affinity for H2Kb (**B**) or H2Kd (**C**) was below a threshold of 500 nM, and whose corresponding wild-type peptides had a binding affinity to the same molecules above 500 nM. These corresponded to Lcn2,

Lect2, and SMAGP. **D, E** C57Bl/6 mice (*n* = 12, 3 per group, data are presented as mean values +/− SEM) were injected intravenously with $10^7$ plaque-forming units (pfu) of VSV-mIFNβ, VSV-*Lcn2*, VSV-*Lect2*, or VSV-*SMAGP*. 10 days later $10^6$ spleno-cytes were co-cultured with a 1:1:1 mixture of SB-HCC explants 1, 2 and 3 at an effector:target (E:T) ratio of 10:1. Supernatants were assayed 48 h later for IL-17 (**D**) or IFNγ (VSV-mIFNβ vs VSV-Lcn2 *p* = 0.0040) (**E**). Significance was determined through ordinary one-way ANOVA with Tukey's multiple comparison tests. Statistical significance was set with * indicating a *p* value less than 0.05, ** <0.01, *** <0.001, and **** <0.0001.

with the highly significant therapy induced by treatment with VSV-SB-HCC1,2,3, SB-HCC tumor-bearing mice were depleted of different immune subsets prior to treatment with VSV-SB-HCC1,2,3 +/− ICI (Fig. 7F). As before, anti-PD-L1 ICI alone generated cures in just under 50% of the mice (Fig. 7F, light blue line) and mice treated with a control

isotype IgG all succumbed to tumor (Fig. 7F, dark blue line). Also, as before, treatment with Early anti-PD-L1 + Late VSV-SB-HCC1,2,3 cured 100% of mice (Fig. 7F, brown line). Treatment with VSV-SB-HCC1,2,3 alone was not significantly different from Early anti-PD-L1 + Late VSV-SB-HCC1,2,3, although one mouse in this experiment succumbed to

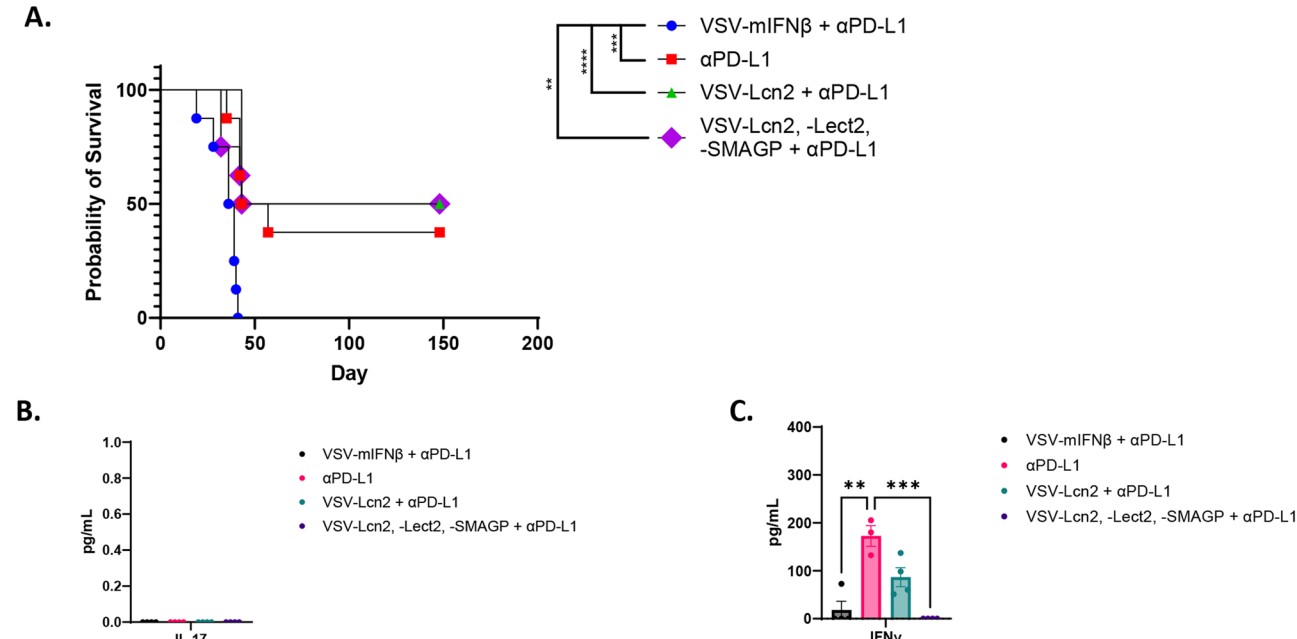

**Fig. 6 | VSV expressing the putative HCC_TAA Lcn2 improves upon therapy with VSV-IFNß in combination with ICI. A** Following hydrodynamic injection of hMet + S45Y ß-Catenin (day 0), mice was treated with anti-PD-L1 (200 μg/injection; days 5, 7, 9, 12, 14, 16) followed by $10^7$pfu of VSV-IFNß, VSV-IFNß-*Lcn2* or with $3 \times 10^6$ pfu each of (VSV-IFNß-*Lect2* + VSV-IFNß-*Lcn2* + VSV-IFNß-*Smagp*) (day 21, 22, 23). $N = 32$. **B, C** In a repeat of the protocol of (**A**), spleens were harvested 10 days following the last dose of virus and $10^6$ splenocytes were co-cultured with SB-HCC

1,2,3 tumor targets at an effector:target ratio of 10:1 with supernatant assayed 48 h later for (**B**) IL-17 and (**C**) IFNγ. $N = 15$, data are presented as mean values +/− SEM, VSV-mIFNβ + αPDL1 vs. αPDL1 $p = 0.0018$, αPDL1 vs. VSV-Lcn2, -Lect2, -SMAGP + α PDL1 $p = 0.0006$. Significance for (**A**) was determined through survival curve comparison testing using a log-rank Mantel-Cox test and (**B, C**) was determined through ordinary one-way ANOVA with Tukey's multiple comparison tests.

tumor (Fig. 7F, black line). Depletion of neither NK, nor CD4⁺, cells from mice treated with the Early anti-PD-L1 + Late VSV-SB-HCC1,2,3 regimen significantly decreased overall therapy, although both groups fell from 100% survival by day 150 (Fig. 7F, red and green lines respectively). However, depletion of CD8⁺ T cells almost completely abolished therapy (Fig. 7F, purple line). Finally, treatment of tumor-bearing mice with Early anti-PD-L1 and Late VSV-cDNA library in which the cDNA was sourced from a melanoma cell line (VSV-ASMEL) was also ineffective (Fig. 7F, orange line), showing that CD8⁺ T cell boosting against cancer type-specific antigens is critical for these effects.

CD8 + T cell-dependent therapy with VSV-SB-HCC1,2,3 +/− ICI was highly associated with IFNγ-mediated anti-tumor effects. Thus, as before, although the anti-tumor T cell response to SB-HCC developed slowly and then waned after about 50 days, anti-PD-L1 ICI maintained the anti-tumor T cell response from d28-d49 and then significantly expanded it through day 90 (Fig. S6A, B). As in Fig. S2, treatment with anti-PD-L1 and VSV-IFNß significantly diminished the strength of the anti-PD-L1-enhanced anti-tumor CD8 + T cell response and replaced it with a potent anti-VSV CD8 + T cell response (Fig. S6C). Treatment with VSV-IFNß-SB-HCC1,2,3 alone (no ICI), which was itself highly therapeutic (Fig. 7C), significantly enhanced the strength of the anti-tumor CD8 + T cell response early (d28 and 49) compared to that of the anti-PD-L1-enhanced CD8 + T cell response (Fig. S6D). Although the magnitude of this response did not fall off at a late time point (day 90), it was not as strong as that induced by anti-PD-L1 ICI alone at day 90 (Fig. S6D). In both cases (early and late) significant anti-tumor CD8 + IFNγ T cell responses co-existed with potent anti-viral (anti-VSV-N_{52-59}) T cell responses (Fig. S6D), which was not the case with treatment with VSV-IFNß (Fig. S6C). Treatment with anti-PD-L1 + VSV-IFNß-SB-HCC1,2,3 induced a significantly enhanced and sustained (day 90) anti-tumor CD8 + T cell response compared to the anti-PD-L1-enhanced response and also co-existed with an anti-VSV CD8 + T cell response (Fig. S6E). These anti-SB-HCC CD8 + T cell responses did not cross

react with murine melanoma cells indicating that the cDNA library approach generates a tumor-specific immune response (Fig. S6E). This was confirmed by the observation that SB-HCC tumor-bearing mice treated with anti-PD-L1 and the ASMEL VSV-cDNA library constructed from melanoma cDNA (as opposed to SB-HCC cDNA) (which was not therapeutic Fig. 7F) significantly inhibited the magnitude of the anti-PD-L1 induced anti SB-HCC tumor CD8 + IFNγ + T cell response, although anti-B16 melanoma CD8 + IFNγ + T cells were abundant (Fig. S6F).

Extensive lysis of SB-HCC1,2,3 live tumor cell targets, but not of B16 melanoma targets, was observed when purified CD8 + IFNγ + T cells from mice treated with anti-PD-L1 ICI were co-cultured in vitro (compared to levels of lysis using CD8 + T cells from untreated SB-HCC- tumor bearing mice) (Fig. S7A, B). Consistent with a loss of both therapy (Fig. 7C) and IFNγ anti-tumor CD8 + T cell response (Figs. S2, S6) in mice treated with both anti-PD-L1 and VSV-IFNß, levels of SB-HCC1,2,3 cell lysis were not significantly different from those of untreated mice (Fig. S7C). In contrast, HCC-specific, CD8 + T cell-mediated lysis of HCC tumor targets was highly significant from mice treated with VSV-SB HCC1,2,3 alone or with VSV-SB HCC1,2,3 plus anti-PD-L1 (Fig. S7D, E). Finally, treatment with the melanoma derived VSV-cDNA ASMEL library plus anti-PD-L1 was not therapeutic against SB-HCC tumors (Fig. 7C), did not induce an HCC-specific IFNγ + CD8 + T cell response (Fig. S6F) and did not generate detectable CD8 + T cell lysis against SB-HCC tumor targets−although a potent CD8 + T cell lysis of murine melanoma cells was observed (Fig. S7F). Taken together, these data show that VSV-IFNß-SB-HCC1,2,3-mediated therapy of SB-HCC tumors correlated very well with generation of IFNγ + CD8 + T cells with strong lytic activity against HCC targets.

Taken together, these data show that optimal therapy requires ICI-induced expansion of anti-HCC_TAA-specific CD8⁺ T cells, which are subsequently further expanded by VSV-IFNß-HCC_TAA boosting, through induction of both Th17 and Th1 component mechanisms.

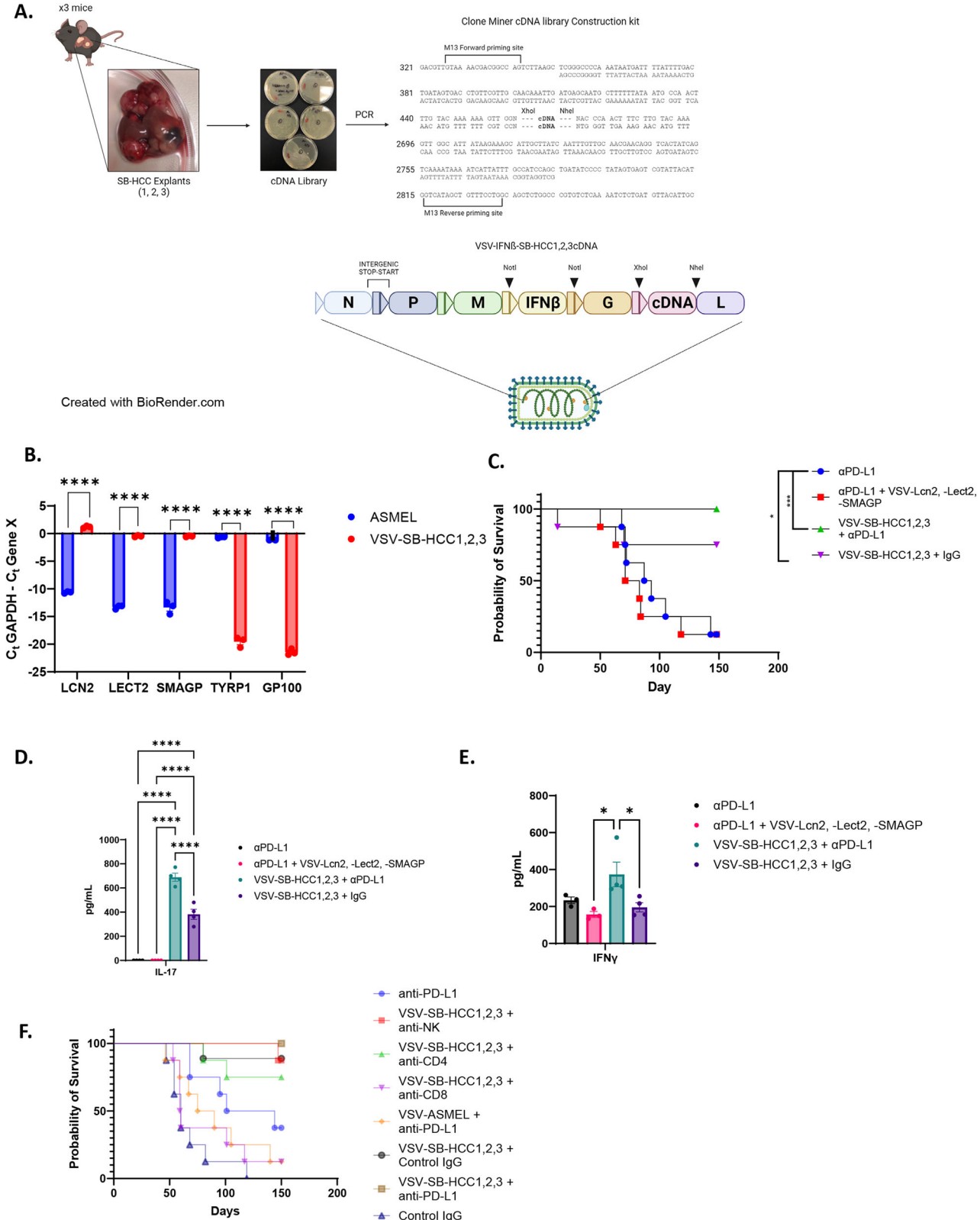

## Discussion

Here, we used a Sleeping Beauty Transposon-based model of HCC which re-capitulates many features of the human disease genetically, phenotypically, and in its partial sensitivity to anti-PD-L1 ICI therapy. We show that, in a tumor model where there is a pre-existing, weak/immuno-subdominant anti-HCC_TAA CD8+ T cell response, addition of a highly immuno-dominant oncolytic virus effectively occluded the anti-

tumor response with a rapid effector anti-viral CD8+ T cell response. However, by encoding a range of HCC_TAA within the virus in the form of an HCC-derived cDNA library, the potent immune adjuvant properties of the virus allowed for boosting of the pre-existing anti-HCC_TAA CD8+ T cells leading to high rates of tumor cures.

With the rationale of disrupting the immune-suppressive TME of HCC and converting it into a virus-inflamed immune infiltrated TME,

**Fig. 7 | VSV-SB-HCC1,2,3 cures mice with SB-HCC mediated by CD8 T cells and associated with Th1 and Th17 responses. A** cDNA from three murine Sleeping Beauty HCC tumor-bearing livers, (SB HCC 1,2&3 Library), was pooled, cloned and amplified by PCR. The Library was then cloned into the VSV backbone plasmid between the *Xho*1 and *Nhe*1 sites. Figure created with BioRender.com under CC BY-NC-ND. **B** Composition of the VSV-derived ASMEL[36] and SB HCC 1,2&3 Libraries was validated using PCR from the equivalent of $10^8$ pfu of the VSV-ASMEL[36] or VSV-SB HCC 1,2&3 with gene specific primers to the HCC-specific *Lcn2*, *Lect2*, and *Smagp* genes or to the melanoma specific *TYRP1* and *GP100* genes. All *p* values noted are <0.0001. **C** Following hydrodynamic injection of hMet + S45Y ß-Catenin (day 0), animals were treated on days 21, 23, 25, 28, 30, 32 with anti-PD-L1 (100 µg/injection) with, or without, $10^7$ pfu of VSV-SB-HCC1,2,3 or $3 \times 10^6$ pfu each of (VSV-IFNß-*Lect2* + VSV-IFNß-*Lcn2* + VSV-IFNß-*Smagp*) on days 38,40,42. (*N* = 32, 8 mice per group). **D, E** $10^6$ splenocytes were co-cultured with a 1:1;1 mixture of live SB-HCC 1,2,3 explant cells as targets at an effector:target ratio of 10:1 with supernatant assayed 48 h later for (**D**) IL-17 and (**E**) IFNγ. (*N* = 12, 3 mice per group, for (**D**) all *p* values noted are <0.0001, for (**E**) αPDL1 + VSV-Lcn2, -Lect2, -SMAGP vs. VSV-SB-HCC1,2,3 + αPDL1 *p* = 0.0230, VSV-SB-HCC1,2,3 + αPDL1 vs. VSV-SB-HCC1,2,3 + IgG *p* = 0.0437) (**F**). Following hydrodynamic injection of hMet + S45Y ß-Catenin (day 0), animals were treated on days 21, 23, 25, 28, 30, 32 with control isotype IgG or with anti-PD-L1 (200 µg/injection) with, or without, PBS or $10^7$pfu ASMEL (VSV-melanoma cDNA library)[36] or $10^7$ pfu VSV-SB-HCC1,2,3 on days 38,40,42. Depleting antibodies were given concurrently with either anti-PD-L1 or IgG isotype control for 6 doses. (*N* = 65, 8 mice per group except VSV-SB-HCC1,2,3 + Control IgG, *n* = 9). Significance comparisons for (Fig. S8) are included in the Supplement, (**B**) was determined using 2-way ANOVA with multiple comparisons using Tukey's multiple comparisons test, (**C, F**) were determined through survival curve comparison testing using a log-rank Mantel-Cox test, and (**D, E**) were determined through ordinary one-way ANOVA with Tukey's multiple comparison tests.

we tested VSV-IFNß gene as an oncolytic and immunotherapeutic agent[15,17–21,24,34] in a clinical trial [NCT01628640] in which VSV-IFNß was administered to patients with sorafenib-refractory HCC. In that study, we observed modest activity associated with VSV-IFNß virus alone. Therefore, we sought to improve clinical utility of the virus in the context of the current standard of care regimen atezolizumab/bevacizumab which includes ICI. We used a slow developing model of HCC, which resembles the etiology of human HCC, in which a Sleeping Beauty transposon integrates the ß-catenin and hMet oncogenes into the livers of neonatal mice resulting in multi-focal tumor formation over 100–150 days (Fig. 1). Tumors grew progressively in the liver, showed characteristics of immune suppression, expressed PD-L1, and were partially susceptible to anti-PD-L1 therapy (~50% long term cures), thereby mimicking several aspects of the human disease and its current therapy (Figs. 1, 2A–C). Based on the data of Figs. S1, S2 and S5, we hypothesize that successful treatment of HCC with anti-PD-L1 treatment must maintain/reach a supra-threshold level of CD8+ anti-tumor T cell reactivity (>~150–200 IFNγ Spots/$10^6$ CD8 + T cells). The reasons why individual mice who are treated with apparently equivalent regimens of ICI either do, or do not, reach this therapeutic threshold of T cell reactivity are unclear; this may be related to purely technical issues associated with treatment administration. However, we believe that it is more likely to be related to the stochastic nature of tumor formation in this model in which hydrodynamic plasmid injections into the liver will be associated with different tumor phenotypes (genetically, antigenically and spatially). In turn, these data are significant in that they suggest the anti-tumor T cell response could be used as an important biomarker of likely patient response to ICI in the clinic.

Contrary to our initial hypothesis, the addition of a highly immunogenic oncolytic virus to a weak, pre-existing anti-PD-L1-sensitive anti-tumor immune response inhibited the therapeutic anti-HCC_{TAA} immune response (Fig. 2D, E) irrespective of treatment sequencing and the combination of virus and ICI therapy was unable to improve on the efficacy of ICI alone. We do not believe that the lack of efficacy of VSV-IFNß alone in this model is due to a lack of oncolytic capacity of the virus. This is because, of multiple cell lines established from tumor explants of SB-HCC tumor-bearing mice, all support replication of, and oncolysis by, VSV-IFNß at high levels. Therefore, we hypothesize that, although spontaneously arising SB-HCC tumors which develop in the context of a fully intact murine immune system, have a highly defective response to Type I IFNs and the potential to support viral replication and oncolysis to a high level, the liver/tumor-bearing liver microenvironment may exert heavily suppressive, tumor cell-extrinsic effects to prevent simple oncolysis from being effective in this model. One possible interpretation of these data is that a highly immuno-dominant anti-viral CD8[+] T cell response induced by VSV-IFNß out competes, or interferes with, the weak, immuno-subdominant, slow developing, anti-PD-L1-sensitive anti-HCC_{TAA} T cell response. Using CyTOF analysis and ELISPOT assays, Fig. 3, S1 and S2 showed that

VSV-IFNß treatment did indeed generate a highly significant expansion of one, or a few, dominant anti-viral effector CD8[+] T cell populations with the concomitant relative disappearance of those putative anti-tumor T cell populations which are the likely target of anti-PD-L1 treatment. The immune profile of the virally induced CD8[+] T cell populations matches that of anti-viral effector CD8[+] T cells, as granzyme B characterizes antiviral CD8[+] T cell responses[49,50]. Other markers also are consistent with this population being an anti-viral effector CD8 population. IL-7R has been shown to be consistently overexpressed on antigen-specific effector CD8[+] T cells during viral infections[51], while Bat3 promotes immune cell proliferation and cytokine production in persistent viral infections by promoting a T helper type 1 response[52,53]. Taken together, these data strongly argue that the abolition of therapeutic benefit achieved with ICI alone by addition of virus to ICI therapy was due to a rapid induction/expansion of anti-viral effector CD8 cells at the expense of anti-tumor, PD-1 expressing terminal effector CD8[+] T cells. Moreover, the repolarization of the effector response of CD8[+] T cells away from anti-HCC_{TAA} CD8[+] T cells towards a population of anti-viral effector CD8[+] T cells suggest that inappropriately timed anti-PD-L1 ICI could actually lead to in vivo enhancement of the immunodominant anti-VSV T cell response over the immuno-subdominant anti-HCC_{TAA} T cell response−expanding anti-viral T cells and not anti-HCC_{TAA} T cells. An implication of these data is that early, prolonged ICI may allow for strengthening/expansion of the immuno-subdominant anti-HCC_{TAA} T cell response with time such that a later immuno-dominant anti-viral T cell response is less able to out-compete it. However, the data of Fig. 2E suggest that even late virus treatment has a dominant negative effect on the anti-tumor T cell response. We also observed a relative reduction in the B cell population in tumor-bearing mice treated with PD-L1 + VSV (Fig. 3B–F). This apparent relative reduction may be attributable to the ability of viral infection to subvert B cell populations in lymphoid organs as previously reported[54]. Alternatively, the large increase in the activated (anti-viral) CD8 + T cell population may contribute to an apparent relative reduction in the B cell population observed in the CyTOF data.

Our previous studies have shown that inclusion of a tumor antigen within VSV induces CD8[+] T cell-dependent anti-tumor therapy directed specifically against the virally encoded antigen as a result of the potent immune-stimulating adjuvant properties of infection with VSV[19,34–36]. Therefore, we hypothesized that it would be possible to amalgamate the potent anti-viral response with an anti-tumor T cell response by expressing immunologically relevant tumor-associated antigens from within the virus. As a model system, we incorporated the OVALBUMIN antigen into the Sleeping Beauty ß-catenin/hMet system such that tumors would express the OVA protein. ß-catenin/hMet/OVA tumors generated spontaneous anti-HCC_{TAA(OVA)} endogenous T cells in both spleen and tumor and addition of anti-PD-L1 ICI expanded these CD8[+] T cell responses (Fig. 4). By encoding the putative HCC_{TAA(OVA)} within oncolytic VSV-ova highly significant numbers of anti HCC_{TAA(OVA)}

T cells were generated in vivo even in the absence of ICI. In addition, those anti-HCC$_{TAA(OVA)}$ CD8$^+$ T cell responses were substrates for expansion by anti-PD-L1 ICI therapy. Moreover, in contrast to the ablation of the anti-HCC$_{TAA}$ CD8$^+$ T cell responses by the addition of VSV virotherapy observed in Figs. 2, 3, combined treatment with virus$_{(TAA)}$ and ICI was able to expand the anti-HCC$_{TAA}$ CD8$^+$ T cell response preferentially over at least one component of the immuno-dominant anti-viral response (Fig. 4). Nevertheless, in this system, OVA represents a highly immunogenic, non-self, non-tolerized HCC$_{TAA}$ and does not reflect the situation in which HCC$_{TAA}$ are likely to be highly immuno-subdominant with low numbers of precursor T cells available to recognize and reject them (Fig. 3). Therefore, using RNAseq analysis of Sleeping Beauty tumors recovered from untreated mice, we identified three separate antigens predicted to be strong MHC Class I binders and, therefore, potential T cell targets when over-expressed in tumors. Taken together, the data of Fig. 5 and S4 show that, of the three potential HCC tumor-associated antigens tested here, only LCN2 is immunogenic. In addition, the weak, slow-developing anti-LCN-2/ HCC CD8 + T cell response was undetectable following low dose ($3 \times 10^6$ pfu) VSV-*Lcn2* treatment at 10d post-infection but expanded by day 30 post-infection. Higher dose treatment ($10^7$ pfu) induced a faster developing, stronger anti-LCN2/HCC response, detectable at both days 10 and 30 post last virus injection. Treatment of SB-HCC tumors with VSV-IFNß-*Lcn2* VSV was still not able to enhance therapy compared to anti-PD-L1 alone, although the presence of Lcn2 did prevent the inhibition of the effects of anti-PD-L1 alone (Fig. 6).

Cumulatively, these data are consistent with a model in which the immuno-dominant anti-VSV CD8$^+$ T cell response ablates the weaker, immuno-subdominant anti-HCC$_{TAA}$ CD8$^+$ T cell response causing loss of the ICI therapy alone when ICI was combined with VSV-IFNß (Fig. 2, 3). However, when at least one potentially immunogenic HCC$_{TAA}$, such as LCN2, was added to VSV-IFNß + anti-PD-L1, early treatment with anti-PD-L1 ICI allowed for expansion of potentially tumor-reactive CD8$^+$ T cells (Cluster 17 in Fig. 3), the anti-LCN2 component of which could then be boosted by late vaccination with VSV-IFNß-*Lcn2*. In contrast, late VSV-IFNß with no additional HCC$_{TAA}$ simply activates a rapid new anti-viral CD8$^+$ T cell population, which replaces/outcompetes the developing, anti-PD-L1-expanded HCC$_{TAA}$-specific CD8$^+$ T cells.

In this model, an endogenous anti-HCC$_{TAA}$ CD8$^+$ T cell response against multiple (weak) HCC$_{TAA}$, expanded by early ICI treatment, would be further boosted by VSV-HCC$_{TAA}$ vaccination. If this model were true, we predicted that by adding multiple further HCC$_{TAA}$ to the vaccinating VSV-IFNß virus, an increased number of anti-HCC$_{TAA}$ T cells could be boosted by late VSV-IFNß-HCC$_{TAA}$ vaccination. In the absence of effective models to predict the nature of multiple possible HCC$_{TAA}$ (Fig. 5), we hypothesized that cloning a cDNA library from Sleeping Beauty HCC tumor lines into VSV would allow for the highly immunogenic display of multiple HCC$_{TAA}$ with the advantage that the HCC$_{TAA}$ would not need to be molecularly identified beforehand to be effective[35,36]. Therefore, cDNA from three separate Sleeping Beauty explants mixed at a 1:1:1 ratio was cloned into the VSV-IFNß virus to give a viral stock of VSV-IFNß-SB-HCC1,2,3cDNA (Fig. 7A). By expressing a multitude of putative, uncharacterized, tumor antigens within VSV, for the first time, a combination of VSV and anti-PD-L1 ICI generated significantly improved therapy over anti-PD-L1 ICI treatment alone with 100% of mice surviving to day 150 (Fig. 7C). Anti-tumor therapy against HCC was not observed if the VSV-cDNA was derived from melanoma instead of HCC tumors, was dependent upon CD8$^+$ T cells but less so on CD4$^+$ T cells and was associated with a Th17 phenotype of T cell activity (Fig. 7). Interestingly, treatment with VSV-IFNß-SB-HCC1,2,3cDNA alone (no anti-PD-L1) was also very effective at generating long term cures. These data suggest that VSV-mediated display of multiple antigens was sufficient of itself to boost slow developing anti HCC$_{TAA}$ CD8$^+$ T cell responses even in the absence of expansion by prior anti-PD-L1 treatment.

A major advantage of using a tumor-derived cDNA library as the source of multiple HCC$_{TAA}$ is that there is no need to define the identity of multiple TAA from each patient. On the other hand, a concern about the use of a cDNA library is that tolerance may be broken to multiple normal liver-associated antigens—although we did not observe any toxicity associated with autoimmune liver disease in the mice cured of their tumors when treated with VSV-IFNß-SB-HCC1,2,3cDNA either with, or without, anti-PD-L1. We believe that by constructing the cDNA library from tumor tissue the representation of normal self-antigens against which any non-tolerized T cells exist in vivo may be reduced relative to tumor associated markers which can form immunological targets. Further studies are underway to assess the full range of toxicities associated with VSV-IFNß-SB-HCC1,2,3cDNA-mediated liver damage.

Our results here provide a cautionary message for the use of highly immunogenic viruses as tumor-specific immune-therapeutics. We show here that oncolytic virotherapy can induce anti-viral T cell responses which can actively inhibit, or obscure, pre-existing weak anti-tumor T cell responses leading, potentially, to decreased anti-tumor therapy with ICI which targets that immuno-subdominant anti-tumor T cell population. However, by ensuring that at least a proportion of the anti-viral T cell response can also act to boost an anti-tumor response by encoding multiple relevant tumor antigens within the virus the CD8$^+$ T cell response associated with oncolytic virotherapy can be purposed for very effective tumor clearance. In the model system used here, oncolytic virotherapy was used in the context of a poorly immunogenic developing tumor against which an endogenous T cell response existed and which could be enhanced by ICI therapy. We are currently investigating the alternative situation in which a developing tumor is completely non-immunogenic—that is it does not raise any anti-tumor T cell response. In such a case it may be that oncolysis by the virus helps to generate a previously non-existent anti-tumor T cell response upon which subsequent ICI treatment may be able to work. Our findings can, therefore, inform the rational sequencing of a combinatorial use of oncolytic virotherapy and ICI therapy such that viral oncolysis is used productively to generate anti-tumor T cell populations upon which immune checkpoint blockade can effectively work.

In summary, our data are consistent with a model in which a poorly immunogenic tumor is partially sensitive to ICI therapy through the selective re-invigoration/expansion of anti-tumor CD8$^+$ T cell populations presumably recognizing weak, sub-dominant antigens. In these circumstances, treatment with an OV encoding a set of highly immunogenic, immuno-dominant viral antigens induces populations of anti-viral CD8$^+$ T cells which overwhelm the pre-existing anti-tumor CD8$^+$ T cell populations and which, therefore, significantly inhibit therapy associated with ICI alone. However, by expressing multiple TAA within the OV, the immune adjuvant properties of the virus become advantageous rather than detrimental by promoting boosting of the pre-existing anti-TAA CD8$^+$ T cell populations in vivo through induction of both Th17 and Th1 component mechanisms. In this way, the anti-viral T cell response is chimerized to become, at least in part, an anti-tumor T cell response leading to a positive interaction between ICI therapy and oncolytic virotherapy.

## Methods

### Pathology
Livers were fixed in 10% formalin and stained with hematoxylin and eosin by the Mayo Clinic Histology Core Facility. Immunofluorescence staining was as described previously[55]. PD-L1 antibody was purchased from BioX-cell, clone 10 F.9G2.

### Cell lines and viruses
VSV expressing murine IFN-ß (VSV-IFN-ß), ovalbumin (VSV-OVA), *Lcn2* (VSV-IFNß-Lcn2), *Lect2* (VSV-IFNß-Lect2), *SMAGP* (VSV-IFNß-SMAGP),

or green fluorescent protein (VSV-GFP) were rescued from the pXN2 cDNA plasmid using the established reverse genetics system in BHK cells as described previously[21,22,34]. In brief, BHK cells were infected with MVA-T7 at an MOI of 1. Cells were incubated at 37 °C and 5% $CO_2$. After 1 h, cells were transfected with pVSV-XN2 genomic VSV plasmid (10 μg), pBluescript (pBS)-encoding VSV-N (3 μg), pBS-encoding VSV-P (5 μg), and pBS-encoding VSV L proteins (1 μg) using Fugene6 according to the manufacturer's recommendations. Cells were incubated at 37 °C and 5% $CO_2$ for 48 h. After 48 h, supernatant was collected and clarified by passing through a 0.2-μm filter. All transgenes were inserted between viral G and L genes using the *Xho*I and *Nhe*I restriction sites with the murine IFN-ß gene cloned between the viral M and G genes. Virus titers were determined by plaque assay on BHK cells. BHK cells were obtained from the ATCC and otherwise not authenticated. VSV-cDNA libraries were generated as described previously[35,36]. Briefly, cDNA from two human melanoma cell lines, Mel624 and Mel888, (ASMEL library)[36] or from three murine Sleeping Beauty HCC tumor-bearing livers, (SB HCC 1,2&3 Library), was pooled, cloned (Clone Miner cDNA Construction kit) (Invitrogen) and amplified by PCR. The PCR-amplified cDNA molecules were size fractionated to below 4 kbp for ligation into the parental VSV genomic plasmid pVSV-XN2[56] between the G and L genes since lower-sized cDNA inserts were associated with both higher viral titers and lower proportions of Defective Interfering particles. The complexity of the ASMEL and SB HCC 1,2&3 cDNA libraries cloned into the VSV backbone plasmid between the *Xho*1 and *Nhe*1 sites was $7.0 \times 10^6$ [36] and $4.1 \times 10^6$ colony forming units respectively (at dilutions of $10^{-6}$ and $10^{-5}$ there were 8 and 74 colonies with the SB HCC 1, 2 & 3 Library). For the SB-HCC1,2,3 Library, of 30 colonies picked at random, 1 had no insert, 12 had an insert of less than 0.5 kbp and 17 had inserts between 0.5 kbp and 4 kbp. Virus was generated from BHK cells by co-transfection of pVSV-XN2-cDNA library DNA along with plasmids encoding viral genes as described[56]. Virus was expanded by a single round of infection of BHK cells and purified by sucrose gradient centrifugation.

## Mice

All experiments utilized 6 to 8-week-old male and female C57Bl/6 mice (stock 000664) obtained from The Jackson Laboratory with the exception of Fig. 1H, which utilized FVB mice (stock 001800) obtained from The Jackson Laboratory. All animals were maintained in a specific pathogen-free BSL2 biohazard facility. Experimental mice were co-housed and exposed to a 12:12-h light-dark cycle with unrestricted access to water and food. The ambient temperature was restricted to 68 °F to 79 °F and the room humidity ranged from 30% to 70%. Animals were euthanized upon reaching any of the following humane endpoints, specifically weight loss ≥20%, inability to ambulate, inability to reach food or water, or a body condition of ≤1 using our IACUC-approved scoring system. Euthanasia was conducted by carbon dioxide inhalation with secondary cervical dislocation. These humane endpoints were not exceeded. All animal studies were conducted in accordance with and approved by the Institutional Animal Care and Use Committee at Mayo Clinic. Sex of the animal was not considered in this study design.

## In vivo experiments

All in vivo studies were approved by the Institutional Animal Care and Use Committee at Mayo Clinic. Our study uses a murine model of human hepatocellular carcinoma (HCC) developed by Dr. Satdarshan Monga in which human Met (hMet) and an activating ß-catenin mutant drive tumorigenesis[33]. Specifically, hMet and ß-catenin point mutants, in this case S45Y, are co-expressed in hepatocytes utilizing the Sleeping Beauty (SB) transposable element system[57]. C57Bl6 mice were injected with pT3-EF5a-hMet-V5 (20 μg), pT3-EF5a- Human ß-catenin with S45Y mutation (20 μg), and pCMV/SB (the hyperactive sleeping beauty expression vector) 1.6 μg (25:1 ratio of oncogenes to SB) diluted in 2 ml of normal saline (0.9% NaCl), filtered through 0.22 mm filter

and injected into the lateral tail vein in 5 to 7 s. For HCC-OVA tumors, an OVA-expressing plasmid, pT3-EF5a-OVA-V5 (20 μg), was added to the pT3-EF5a-hMet-V5 (20 μg), pT3-EF5a- Human ß catenin with S45Y mutation (20 μg), and pCMV/SB 1.6 μg plasmids diluted in 2 ml of normal saline prior to hydrodynamic injection.

Following hydrodynamic injection, animals were treated as described in the text. Specific antibody doses were: anti-mouse PD-L1 (200 or 100 μg per injection depending on the experiment, please see text) (BioX-Cell, clone 10 F.9G2), CD8α (200 μg/mouse) (BioX-Cell, clone 2.43), CD4 (200 μg/mouse) (BioX-Cell, clone GK1.5), NK1.1 (200 μg/mouse) (BioX-Cell, clone PK136), and IgG2A isotype (200 μg/ mouse) (BioX-Cell, clone 2A3). All antibody treatments were given i.v. by tail vein unless otherwise specified.

In vivo treatment of SB-HCC tumor-bearing mice with VSV-IFNß was given by injection into the hepatic artery typically at $10^7$ pfu/dose delivered in 50 μl volume (please see text for individual dosings). Therefore, virus was delivered equally across the tumor-bearing livers rather than into specific lesions. Control mice were subjected to the same surgical interventions for injection of PBS or VSV-GFP. Other viruses (VSV-Lcn2, VSV-Lect2, or VSV-SMAGP) were given by the same route at either $10^7$ pfu/dose (VSV-Lcn2) or at $3 \times 10^6$ pfu/dose (combination of VSV-Lcn2, VSV-Lect2 and VSV-SMAGP).

## Immune cell isolation

Spleens from C57Bl/6 mice were immediately excised upon euthanasia. Single-cell suspensions were generated in vitro via mechanical dissociation. Red blood cells were lysed by resuspension in ammonium-chloride-potassium lysis buffer and incubating at room temperature for 2 min.

Mice were euthanized and livers were perfused with 20 mL PBS before being removed. Excised livers were dissected and passed through a 200-gauge stainless steel mesh. Cells were re-suspended in RPMI 1640/FCS. Cells were centrifuged (1500 r/min) and the resulting pellet resuspended in 40% Percoll solution with heparin. Cells were loaded on a 70% Percoll solution and centrifuged at 2000 r/min for 20 min at room temperature. Cells recovered from the gradient were centrifuged and washed with HBSS/FCS.

## Flow cytometry/CyTOF analysis

Tumor-infiltrating lymphocytes were isolated as above in Flow Cytometry staining buffer. Immune cells were analyzed using FlowJo 10. Mouse cells were stained with fluorochrome-conjugated antibodies against combinations of the following antigens: CD3 (Biolegend # 100236 clone 145-2C11, dilution 1:500), CD8a (Biolegend #100738/ 100747, clone 53-6.7, dilution 1:1000), CD4 (Biolegend #100451 clone GK1.5), PD1 (Biolegend # 109110, clone RMP1-30, dilution 1:200), TIM3 (Biolegend #119704, clone RMT3-23, dilution 1:200), CD11c (BD Biosciences clone HL3), I-A/I-E (MHCII) (Biolegend M5/114.15.2), CD86 (Biolegend #105037, clone GL-1), and fixable live dead viability dye (Zombie NIR). Cells were stained with the H-2Kb VSV NP$_{52-59}$ RGY-VYQGL (Brilliant Violet 421–labeled) tetramer at a dilution of 1:500 or the H-2Kb chicken ova$_{257-264}$ SIINFEKL (APC-labeled) tetramer at a dilution of 1:150, which were obtained from the National Institutes of Health Tetramer Core Facility.

CyTOF analysis was conducted through the Mayo Clinic CyTOF Core using a 44-label combined T cell panel. CyTOF analysis was conducted using RStudio and Cytofkit, with individual populations isolated using Rphenograph and FlowSOM tSNE analyses for 22 and 9 populations, respectively.

## ELISA

Following isolation, splenocytes were resuspended at a concentration of $1 \times 10^6$ cells/mL in Iscove's modified Dulbecco's medium (Gibco) supplemented with 5% FBS, 1% penicillin-streptomycin, and 40 μmol/L 2-mercaptoethanol. Splenocytes were re-stimulated with $10^5$ live

tumor cells which were a 1:1:1 mix of three different sleeping beauty tumor cell lines recovered from untreated mice at euthanasia (between 75 and 100 days post hydrodynamic injection). Supernatants were collected and assayed for IL-17 and IFNγ by enzyme-linked immuno-sorbent assay (ELISA) as per the manufacturer's instructions (Mouse IL-17 or Mouse IFNγ ELISA Kit, OptEIA, BD Biosciences).

## RNA-sequencing analysis of gene expression

hMet + S45Y ß-Catenin Sleeping Beauty transposon system liver tumor-induced mice sacrificed on day 18 with livers harvested for RNA extraction (Rneasy, Qiagen). RNA samples were then subjected to RNA-seq analysis at the Genome Analysis Core, Mayo Clinic. The 10 most-expressed genes identified and full-length sequences were filtered through NET MHC 2.0 binding affinity algorithm to identify octamer or nonamer peptides whose binding affinity for H2K$^b$ or H2K$^d$ was below a threshold of 500 nM, and whose corresponding wild-type peptides had a binding affinity to the same molecules above 500 nM. A list of 10 candidates was further refined using the EMBL-EBI Expression Atlas for expression in liver tissue to identify 4 high-affinity peptides. This methodology is discussed in detail in our previous work[58].

## PCR validation of ASMEL and SB-HCC1,2&3 libraries

RNA was prepared from the equivalent of 10$^8$ pfu of the VSV-based ASMEL[36] or SB-HCC1,2&3 Library with the QIAGEN RNA extraction kit. 1 µg total RNA was reverse transcribed in a 20 µl volume using oligo-(dT) as a primer. A cDNA equivalent of 1 ng RNA was amplified by PCR with gene specific primers to the HCC-specific *Lcn2*, *Lect2*, and *Smagp*, or melanoma specific *TYRP1* and *GP100* genes.

## Statistical analysis

All analysis was performed within GraphPad Prism software (Graph-Pad). Multiple comparisons were analyzed using one- or two-way analysis of variances with a Tukey's post hoc multi-comparisons test. Survival data were assessed using a log rank Mantel-Cox test. Data are expressed as group mean ± SEM unless otherwise stated. Statistical significance was set with * indicating a p value less than 0.05, ** <0.01, *** <0.001, and **** <0.0001 unless otherwise stated.

All experiments were designed, powered and analyzed by consultation with the Mayo Clinic Bioinformatics and Statistical Core. Many studies included "control" groups important to verify that the animal study was performed correctly, but are not of biological interest. For all analyses, α = 0.05 were used. Assuming σ = 30, 10 mice/grp will provide 80% power to detect a difference of 40 d, with a two-sided α = 0.05. Alternatively, assuming σ = 6 we would be able to detect a difference of 10 d. Large effect sizes were expected and we have been able to detect statistically-significant changes with 7–8/group, highlighting the feasibility of these studies.

## Statistical and reproducibility

Experiments noted in Figs. 1, 3, 5, and 7B were replicated with similar results once.

Experiments noted in Figs. 2, 4, 6, and 7C–F were replicated with similar results at least two times. All experiments were designed, powered and analyzed by consultation with the Mayo Clinic Bioinfor-matics and Statistical Core. Many studies included "control" groups important to verify that the animal study was performed correctly, but are not of biological interest. For all analyses, α = 0.05 were used. Assuming σ = 30, 10 mice/grp will provide 80% power to detect a dif-ference of 40d, with a two-sided α = 0.05. Alternatively, assuming σ = 6 we would be able to detect a difference of 10 d. Large effect sizes were expected and we have been able to detect statistically-significant changes with 7–8/group, highlighting the feasibility of these studies. No data was excluded. For in vivo studies, mice were randomized at time of tumor implantation using the GraphPad QuickCalcs online tool (https://www.graphpad.com/quickcalcs/randMenu/). For in vitro studies, no randomization was performed as cell used in this study were pulled from a single preparation with no reason to believe that the spacial location in the well impacted results. Mice were assessed by a single-blinded individual. For in vitro studies the investigators were not blinded to the allocation of groups during experiments or subse-quently during the analysis. Fully blinded experiments were not pos-sible due to personnel availability to accommodate such situations.

## Reporting summary

Further information on research design is available in the Nature Portfolio Reporting Summary linked to this article.

## Data availability

The data generated in this study have been deposited in a publically available figshare database, https://doi.org/10.6084/m9.figshare.25651704.v1. Data not published or found within the figshare data-base will be made available by request from any qualified investigator. There is no custom code or algorithms used in this study. Source data are provided with this paper.

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

## Acknowledgements

The authors thank Kevin Pavelko for expertise and assistance with CyTOF analysis and Toni L. Woltman for expert secretarial assistance.

Funding was provided by the National Institutes of Health R21 CA262994(RV); R01 AI170535-01(RV); R01 CA269384-01(RV); P50 CA210964-05(RV), The Richard M. Schulze Family Foundation(RV), the Mayo Foundation(RV), the Shannon O'Hara Foundation(RV), Hyundai Hope on Wheels(RV), and a University of Minnesota and Mayo Partnership Award(RV).

## Author contributions

Conceptualization: M.J.W., J.v.V., L.E., T.S., B.K., R.V. Data Acquisition: M.J.W., J.v.V., L.E., T.S., B.K., Jason T, Jill T, Muriel M, Madelyn M, M.P.C.Y., R.V. Data Analysis: M.J.W., J.v.V., L.E., T.S., B.K., R.V. Funding Acquisition: L.E., S.P.S.M., Mark M, M.J.B., L.R.R., R.V. Methodology: M.J.W., J.v.V., L.E., T.S., B.K., Jason T, Jill T. Project Administration: S.P.S.M., Mark M, M.J.B., L.R.R., R.V. Manuscript Writing: M.J.W., R.V. Manuscript Review: M.J.W., J.V.V., L.E., T.S., Jason T, Jill T, Muriel M, Madelyn M, M.P.C.Y., M.O., A.B., A.M., S.P.S.M., Mark M, M.B., L.R.R., RV.

## Competing interests

The authors declare no competing interests.

## Ethics

All researchers fulfill all criteria outlined by the Global Code of Conduct for Research in Resource-Poor Settings.

## Additional information

[1]Department of Hematology/Medical Oncology, Mayo Clinic, Rochester, MN 55905, USA. [2]Department of Molecular Medicine, Mayo Clinic, Rochester, MN 55905, USA. [3]Department of Medical Genetics, University of British Columbia, Vancouver, BC V5Z1L3, Canada. [4]Michael Smith Genome Sciences Department, BC Cancer Research Institute, Vancouver, BC V5Z1L3, Canada. [5]Department of Pharmacology, University of Minnesota, Minneapolis, MN 55455, USA. [6]Division of Pediatric Hematology and Oncology, University of Minnesota, Minneapolis, MN 55455, USA. [7]Department of Veterinary Clinical Sciences, University of Minnesota, St. Paul, MN 55108, USA. [8]Masonic Cancer Center, University of Minnesota, Minneapolis, MN 55455, USA. [9]Clinical Investigation Center, University of Minnesota, St. Paul, MN 55108, USA. [10]Mayo Center for Biomedical Discovery, Mayo Clinic, Rochester, MN 55905, USA. [11]Pittsburgh Liver Institute, University of Pittsburgh and UPMC, Pittsburgh, PA 15261, USA. [12]Department of Hematology/Medical Oncology, Mayo Clinic, Phoenix, AZ 85054, USA. [13]Division of Radiotherapy and Imaging, Institute of Cancer Research, Chester Beatty Laboratories, London SW3 6JB, UK. [14]Department of Gastroenterology and Hepatology, Mayo Clinic, Rochester, MN 55905, USA. [15]Department of Immunology, Mayo Clinic, Rochester, MN 55905, USA. [16]Joan Reece Department of Immuno-oncology, King's College London, London, UK. ✉e-mail: vile.richard@mayo.edu

