## [Peer Review File · Nature Communications]

Expression of tumor antigens within an oncolytic virus enhances the anti-tumor T cell responseREVIEWER COMMENTS

Reviewer #1 (Remarks to the Author):

In this manuscript, Webb et al study the interesting and important topic of efficiency of oncolytic viruses as combined with checkpoint therapy in the context of murine HCC. Design of combination therapies to improve the efficiency and efficacy of checkpoint therapy is important and needed. The authors combine anti-viral and anti-tumor immune responses by encoding tumor antigens within a virus in a murine model of HCC. Although the topic is important and the findings are interesting, there are major points that need to be addressed.

1- Using CyTOF, the authors show expansion of a CD8+ T cell population that they call "putative anti-viral" with characterization of a few markers (Figure 3). The "virus-specificity" of this population needs to be addressed and demonstrated. In addition, the authors ignore the expansion of a large population of B cells in the HCC + anti-PDL1 + VSV which needs to be addressed and discussed.

2- How come there are no Tregs in the tumor bearing group in Figure 3? If there are tregs, they need to be labeled, although the heat map does not show any clusters that can be labeled as such. If there are Tregs present, the authors need to discuss the changes if any in the different groups?

3- The IFN-g readout is confusing and does not correlate with the survival seen in Figure 6A. The authors suggest that the responses are TH1/IFNg dependent. However, the IFN-g measurement shows that although the VSV-Lcn2, Lect2, -SMAGP + α PD-L1 shows similar survival pattern as VSV-LCN, there is no IFNG-production in this group. The authors need to address this and discuss it. Is the survival seen in Figure 6, due to Th1 responses or not?

4- How do the authors explain the discrepancy of IFN-g production between VSV-Lcn2, -Lect2, -SMAGP + α PD-L1 group in Figure 6C and Figure 7E.

5- The mechanism behind the survival is not clear. Is this a IFNg dependent mechanism and are the cell lytic. Why is the function limited to measurement of IL-17 and IFNg. The authors need to do a comprehensive cytokine measurement as well as lytic function of the T cells.

6- The authors need to study the liver microenvironment upon vaccination with the different groups. This should be done after the finish of the treatments (After day 30-32). The composition and phenotype of the cells needs to be shown by flow cytometry.

Minor points:

1- The method described for isolation of cells from liver tissue is incomplete and will not lead to high purity of cells. The authors need to clarify if this is the methodology used or if further purification of the cells is missing from the Methods.

Reviewer #2 (Remarks to the Author):

The concept of combining oncolytic viruses with ICIs as a synergistic immunotherapy approach in solid cancers has been under intense investigation. Many data have shown that an immunogenic OV can serve to heat up the tumor microenvironment and thereby sensitize it to immune checkpoint inhibition. In the submitted work, Webb et al highlight the risk that the use of highly immunogenic OVs could actually be counter-productive to the combination approach, as it could distract the cytotoxic T cell response away from an anti-tumoral and toward an anti-viral one. They further go on to show that this effect can be exploited by using the virus as a vaccine to express dominant tumor associated antigens, which could potentially redirect the immune response back

toward the tumor and then synergize with ICI therapy. The important take-home message here is that OV-based combination approaches with ICIs must be carefully characterized and optimized in order to avoid counteractive effects. Although this work contains novel and important findings, there are a number of signification concerns which should be addressed prior to consideration of publication.

1. There is generally a lack of detail in terms of the dosing and set-up of in vivo experiments. In the methods section there is no mention at all of use of virus in vivo, and the doses can often only be found in some of the figure legends. Doses of antibodies, particularly the immune cell-depleting antibodies are not provided at all. Most importantly, the critical information regarding the administration of virus is missing. It is mentioned once in the results for Figure 2 that the virus was injected IT. Was this the case for all experiments? If so, what exactly is the procedure? Are the mice subjected to laproscopic procedure to inject into the tumors? If so, are the control mice also subjected to the same surgical intervention? Very importantly, is the virus injected into a single lesion or divided among multiple lesions? What volume is applied? Can any distinction be made in therapeutic effects in directly injected versus uninjected lesions?

2. It was mentioned that the maximum tolerated dose of VSV-IFN β in patients was determined to be 3×10^6 TCID $_{50}$, while the dose given in the described mouse studies was 107 TCID $_{50}$. Can the authors comment on this large discrepancy in dosing, especially considering the massive difference in body weight of a mouse vs. human?

3. Why was the vector encoding IFN β used for these experiments? Does IFN β exert any additional therapeutic effects in this model, or is this used simply for safety to reduce neurotoxic effects of VSV? Is it possible that the expression of IFN β is partially responsible for the loss of efficacy of anti-PD-L1 when administered in combination? Although differences in T cell responses of VSV-IFN β versus VSV-GFP are provided in Figure 4, it would be interesting to see how these vectors compare in terms of survival prolongation.

4. Immune cells are characterized in several experiments in what is referred to as "liver tissue". However, it is not explained whether this is a combination of tumor and liver tissue. Ideally, carefully isolated tumor tissue with minimal liver margins should be used for the analysis.

5. How is the development of tumor confirmed prior to the start of treatment and how are mice stratified to treatment groups? It is presumed that tumor development is not 100% in this model and that there is a substantial amount of heterogeneity in terms of tumor progression. Ideally, the mice should be imaged prior to the start of treatment, and mice harboring tumors of a predefined size/number should be randomized to treatment groups such that average tumor burden is roughly equal among groups.

6. It is stated that treatment was sometimes initiated on day 21, although the authors only show a first histological analysis of the liver on day 28, at which time tumors appear to be at a quite early stage of progression. Are tumors already evident on day 21? And is this a physiologically relevant time-point to initiate treatment considering the clinical presentation of the disease?

7. In the experimental set-up of the survival study shown in Figure 2d, anti-PD-L1 therapy is initiated on day 28, as opposed to day 21 in the previous experiments. It looks as if anti-PD-L1 therapy was no longer statistically significant in terms of survival prolongation in this treatment scheme (there are some long term survivors, but the non-survivors die at the same time as control mice). Therefore, the conclusion that the combination with VSV-IFN β abrogates the anti-PD-L1 treatment seems to be overreaching. Are differences in survival prolongation of anti-PD-L1 versus VSV-IFN β + anti-PD-L1 therapies statistically significant in this experiment?

8. It is generally clear the oncolytic virus therapy has no therapeutic effect as a monotherapy in this tumor model. Can the authors comment on why that is the case? Could synergism of OV + ICI therapy be achieved if a more pronounced oncolytic effect could be produced?

9. In the experiments described in Figure 4, it seems that the experiment utilizing VSV-OVA employed a different dosing scheme than the experiments with VSV-GFP or VSV-IFN β . Therefore,

it is difficult to compare the results. Similarly, the authors ask us to compare Figures 4G/H with Figures 4A/B (line 280), but it seems that the time-point for analysis was different (day 23 vs. 98). Therefore, this this does not seem to be a fair comparison.

10. In Figure 4G/H, it is indicated that the tissue was harvested on day 98. However, if we look at the survival curves for this model, it seems that control mice begin dying at around 30 days post-induction, and most mice are already dead by day 98. Is analysis of tissue on day 98 not then inherently biased for the responding/surviving mice?

11. In the survival curves depicted in Figures 6a and 7c, it is interesting that the mice diverge into two vastly different response phenotypes: either they die early (overlapping with control mice) or they have complete responses. Can the authors offer any explanation for this phenomenon? It is also difficult to see how these survival differences can be statistically significant compared to controls when the median 50% survival appears nearly identical.

12. In the experiments in which TAA-expressing vectors were used, it would have been helpful to characterize if/how the T cell responses were shifted back from anti-viral to anti-tumoral.

13. Why is the low-dose anti-PD-L1 therapy seemingly effective in Figure 7f but not in 7c? In general, it is difficult to assess relative efficacy without the PBS or IgG controls.

14. Is this tumor model prone to metastasis? If so, do the mice die from the primary tumors or from the metastases? This would be important to discriminate due to the OV therapy being administered locally to the hepatic tumors.

Additional technical comments:

1. The anti-PD-L1 IHC images shown in Figure 1 are not clear and are difficult to draw conclusions from.
2. The images in 7A are rather small and difficult to see.

RESPONSE TO REVIEWERS – NCOMMS-23-53622

Reviewer #1:

1- Using CyTOF, the authors show expansion of a CD8+ T cell population that they call “putative anti-viral” with characterization of a few markers (Figure 3). The “ virus-specificity” of this population needs to be addressed and demonstrated.

We initially included a tetramer to the immune-dominant epitope of VSV N protein – N₅₂₋₅₉ -in our CyTOF panel to address the Reviewer’s point here. However, the VSV-N₅₂₋₅₉ tetramer did not work well in the context of CyTOF which is why we did not include it in the reported data. However, to address the Reviewer’s point we have now added new flow cytometry data in the new **Figure S1** and in the following text (Page 7, line 183):

To characterize the dynamics of the anti-tumor CD8+ T cell response in more detail, **Figure S1** shows that untreated mice bearing Sleeping Beauty HCC tumors developed CD8+ T cell responses against tumor explants which increased in magnitude with time (day 0 to day 28 following tumor initiation) using ELISPOT analysis. However, between day 28 and 49 these anti-SB-HCC CD8+ T cells responses waned significantly and by day 90 – at which time point most of the control mice have succumbed to the tumor – the anti-SB-HCC tumor CD8+ T cell response was low in the single surviving animal (**Figure S1.A**). When SB-HCC tumor-bearing mice were treated with anti-PD-L1 immune checkpoint inhibition, anti SB-HCC CD8+ T cell responses were maintained in strength and even slightly increased through day 49 post tumor initiation. In addition, treatment with anti-PD-L1 ICI significantly enhanced the magnitude of the CD8+ T cell response by day 90 which was associated with the prolonged survival of mice to this time point (**Figure S1.B**).

And a new **Figure S2** and in the following text (Page 8, line 213):

To define the mechanistic basis of the inhibition of anti-PD-L1 therapy by virus we used ELISOT assays to follow the kinetics of the anti-tumor CD8+ T cell response. As before (**Figure S1.A**), untreated SB-HCC-bearing mice developed a weak, but detectable, anti-SB-HCC CD8+ T cell response with time (day 0-day 28) which significantly declined by days 49 and 52 at which time mice were succumbing to disease (**Figure S2.A**). However, treatment with VSV-IFN β was accompanied by both a significant loss of the anti-SB-HCC CD8+ T cell response by ELISPOT reactivity to live SB-HCC explanted tumor cells and the acquisition of a potent antiviral CD8+ T cell response as measured by ELISPOT reactivity to the immunodominant VSV-N₅₂₋₅₉ epitope (**Figure S2.B**). These data suggest that treatment with oncolytic VSV-IFN β replaces a weak, slowly developing anti-tumor CD8+ T cell response with a much stronger anti-viral CD8+ T cell response. Similarly, as before (**Figure S1.B**), SB-HCC tumor bearing mice treated with anti-PD-L1 ICB developed a stronger, more prolonged anti-tumor CD8+ T cell response than in the absence of ICI (**Figure S2.C**). Moreover, these high levels of anti-tumor CD8+ T cell responses against SB-HCC were associated with significantly longer survival (mice surviving to day 90) (**Figure S2.C**). Consistent with the loss of therapy that we observed

when anti-PD-L1 was combined with VSV-IFN β (**Figures 2D,E**), the anti-SB-HCC CD8+ T cell response observed with both anti-PD-L1 and VSV-IFN β was significantly inhibited relative to anti-PD-L1 ICB treatment alone (**Figures S2.C vs D**) in mice which succumbed to disease at early time points (days 57, 61 and 90). Loss of these anti-tumor CD8+ T cell responses with combined anti-PD-L1 and VSV-IFN β treatment was associated with the generation of potent anti-VSV CD8+ T cell responses (**Figure S2.D**). These data suggest that treatment with oncolytic VSV-IFN β significantly inhibits the ICI-strengthened, therapeutic, anti-tumor CD8+ T cell response by competition with a much stronger anti-viral CD8+ T cell response.

And to the **Results** (page 11, line 291):

That these cells were anti-viral was confirmed using ELISPOT reactivity to the immune-dominant NSV-N₅₂₋₅₉ epitope in **Figures S2 B&D**.

In addition, the authors ignore the expansion of a large population of B cells in the HCC + anti-PDL1 + VSV which needs to be addressed and discussed.

We are unclear as to the Reviewer's point here. The CyTOF data in **Figure 3** shows a heavy relative *decline* in the B cell population in the HCC + anti-PDL1 + VSV group rather than expansion of a large population of B cells. The large expansion of the population of activated CD8+ T cells (for which we have now added data to confirm are anti-VSV T cells) makes it difficult to assess the true numbers of B cells because the CyTOF data provides relative abundance comparisons. To address the change in the B cell population referred to by the Reviewer (but as a decline rather than as an expansion) we have added the following text on page 21, line 617 to the **Discussion**:

We also observed a relative reduction in the B cell population in tumor bearing mice treated with PD-L1+VSV (**Figure 3B-F**). This apparent relative reduction may be attributable to the ability of viral infection to subvert B cell populations in lymphoid organs as previously reported⁵⁴. Alternatively, the large increase in the activated (anti-viral) CD8+ T cell population may contribute to an apparent relative reduction in the B cell population observed in the CyTOF data.

2- How come there are no Tregs in the tumor bearing group in Figure 3? If there are tregs, they need to be labeled, although the heat map does not show any clusters that can be labeled as such. If there are Tregs present, the authors need to discuss the changes if any in the different groups?

To address the Reviewer's point here, we have added the following text to the **Results** on page 10, line 267:

Although the markers FoxP3, CD25/IL2r and CD127/IL7R were included in the CyTOF panel, we were unable to identify predominant populations of Treg within the data set from these SB-HCC tumors (**Figure 3**). Since the CyTOF data represents only relative abundances of different lymphocyte populations it is not possible to exclude the presence

of a functionally important Treg population within these tumors and further detailed studies using other methods are underway to characterize these cell types.

3- The IFN-g readout is confusing and does not correlate with the survival seen in Figure 6A. The authors suggest that the responses are TH1/IFNg dependent. However, the IFN-g measurement shows that although the VSV-Lcn2,Lect2, -SMAGP + α PD-L1 shows similar survival pattern as VSV-LCN, there is no IFNG-production in this group. The authors need to address this and discuss it. Is the survival seen in Figure 6, due to Th1 responses or not?

To address the Reviewer's point here, we have added new data as a new **Figure S4** which dissect the immune responses to each of the LCN2, Lect2 and SMAGP proteins in more detail. Therefore, we have added the following data/text to the **Results** one page 14, line 398:

To explain these observations, we measured the CD8+ T cell responses to each of these antigens in more detail. In **Figure 6C**, 10 days following the last dose of VSV-Lcn2 at dose of 3×10^6 pfu (as part of the virus combination group) the anti-SB-HCC tumor response was undetectable (<20 IFN γ spots/ 10^6 CD8+ T cells/48hrs) and not different from that induced by VSV-GFP (**Figure S4.A**). In contrast, the higher dose of 10^7 pfu of VSV-Lcn2 (single virus treatment) induced a significantly higher anti-tumor CD8+ T cell response (>20 IFN γ spots/ 10^6 CD8+ T cells/48hrs) at the 10day time point (**Figure S4.B**). Therefore, we believe that the difference in the anti-tumor CD8+ T cell responses between the virus combination (lack of IFN γ) and single VSV-Lcn2 virus treatment (low but detectable IFN γ) groups in **Figure 6C** is a consequence of a dose response to VSV-Lcn2 (3×10^6 pfu in the combination virus treatment group compared to 10^7 pfu in the single virus treatment group). Treatment with neither VSV-smagp nor VSV-Lect2 induced anti-tumor CD8+ T cell responses in excess of those induced by the non-specific T cell activation seen with VSV-GFP at either low or high dose of viruses (**Figures S4. A&B**).

4- How do the authors explain the discrepancy of IFN-g production between VSV-Lcn2, -Lect2, -SMAGP + α PD-L1 group in Figure 6C and Figure 7E.

In **Figure 6C**, mice were treated with 3×10^6 pfu each of (VSV-IFN β -Lect2 + VSV-IFN β -Lcn2 + VSV-IFN β -Smagp) and spleens were harvested 10 days following the last dose of virus and IFN γ measured in response to SB-HCC 1,2,3 target cells. In **Figure 7E** mice were treated with 3×10^6 pfu each of (VSV-IFN β -Lect2 + VSV-IFN β -Lcn2 + VSV-IFN β -Smagp) and spleens were harvested at ethanasia (>50days post virus) rather than 10 days post virus in **Figure 6C**.

To explain the apparent discrepancy of IFN γ production between VSV-Lcn2, -Lect2, -SMAGP + α PD-L1 group in **Figure 6C** and **Figure 7E**, we have added new data as a new **Figure S4** and the following text to the **Results** on page 17, line 471:

We also observed an apparent discrepancy between the lack of an anti-SB-HCC IFN γ response in mice treated with (VSV-IFN β -Lect2 + VSV-IFN β -Lcn2 + VSV-IFN β -Smagp) in **Figure 6C** and the detectable anti-SB-HCC IFN γ response in **Figure 7E**. This can be

explained, we believe, because at day 30 after the last injection of virus the anti-tumor CD8+ T cell response induced by VSV-*Lcn2* alone, at both low (3×10^6 pfu) and high (10^7 pfu) doses, was significantly higher than at 10 days post the last injection of virus (**Figures S4 A&B**). Therefore, the slow developing anti-*Lcn2* CD8+ T cells response (**Figure S4.A&B**) was not detectable early (10 days) post virus treatment (**Figure 6C**) but was clearly detectable late (>30 days when mice succumbed to tumor in **Figure 7E**) after the last virus injection.

And we have added the following text to the **Discussion** on page 22, line 646:

Taken together, the data of **Figures 5** and **S4** show that, of the three potential HCC tumor associated antigens tested here, only LCN2 is immunogenic. In addition, the weak, slow developing anti-LCN-2/HCC CD8+ T cell response was undetectable following low dose (3×10^6 pfu) VSV-*Lcn2* treatment at 10d post infection but expanded by day 30 post infection. Higher dose treatment (10^7 pfu) induced a faster developing, stronger anti-LCN2/HCC response, detectable at both days 10 and 30 post last virus injection.

5- The mechanism behind the survival is not clear. Is this a IFN γ dependent mechanism and are the cell lytic. Why is the function limited to measurement of IL-17 and IFN γ . The authors need to do a comprehensive cytokine measurement as well as lytic function of the T cells.

To elaborate on the IFN γ -dependent mechanisms associated with increased survival of mice in **Figure 7** as requested by the Reviewer, we have added new data:

- 1). As a new **Figure S6** showing that mice which survive following treatment with the VSV-SB-HCC1,2,3 cDNA library plus anti-PD-L1 ICB have populations of IFN γ -secreting anti-tumor CD8+ T cells and
- 2). As a new **Figure S7**, showing that those CD8+ T cells have highly lytic against HCC tumor cells, but not against B16 melanomas.

Therefore, we have added new text and data to the **Results** on page 18, line 497:

CD8+ T cell dependent therapy with VSV-SB-HCC1,2,3 +/- ICI was highly associated with IFN γ -mediated anti-tumor effects. Thus, as before, although the anti-tumor T cell response to SB-HCC developed slowly and then waned after about 50 days, anti-PD-L1 ICI maintained the anti-tumor T cell response from d28-d49 and then significantly expanded it through day 90 (**Figures S6 A&B**). As in **Figure S2**, treatment with anti-PD-L1 and VSV-IFN β significantly diminished the strength of the anti-PD-L1-enhanced anti-tumor CD8+ T cell response and replaced it with a potent anti-VSV CD8+T cell response (**Figure S6 C**). Treatment with VSV-IFN β -SB-HCC1,2,3 alone (no ICI), which was itself highly therapeutic (**Figure 7C**), significantly enhanced the strength of the anti-tumor CD8+ T cell response early (d28 and 49) compared to that of the anti-PD-L1-enhanced CD8+ T cell response (**Figure S6 D**). Although the magnitude of this response did not fall off at a late timepoint (day 90), it was not as strong as that induced by anti-PD-L1 ICI alone at day 90 (**Figure S6 D**). In both cases (early and late) significant anti-tumor CD8+ IFN γ T cell responses co-existed with potent anti-viral (anti-VSV-N₅₂₋₅₉) T cell responses (**Figure S6 D**), which was not the case with treatment with VSV-IFN β (**Figure S6 C**).

Treatment with anti-PD-L1+VSV-IFN β -SB-HCC1,2,3 induced a significantly enhanced and sustained (day 90) anti-tumor CD8 $^+$ T cell response compared to the anti-PD-L1-enhanced response and also co-existed with an anti-VSV CD8 $^+$ T cell response (**Figure S6 E**). These anti-SB-HCC CD8 $^+$ T cell responses did not cross react with murine melanoma cells indicating that the cDNA library approach generates a tumor specific immune response (**Figure S6 E**). This was confirmed by the observation that SB-HCC tumor bearing mice treated with anti-PD-L1 and the ASMEL VSV-cDNA library constructed from melanoma cDNA (as opposed to SB-HCC cDNA) (which was not therapeutic **Figure 7F**) significantly inhibited the magnitude of the anti-PD-L1 induced anti SB-HCC tumor CD8 $^+$ IFN γ $^+$ T cell response, although anti-B16 melanoma CD8 $^+$ IFN γ $^+$ T cells were abundant (**Figure S6 F**).

In response to the Reviewer's question about whether the CD8 $^+$ IFN γ T cells are lytic to their targets, we have added new data as a new **Figure S7**. Therefore, we have added the following text to the **Results** on page 18, line 523:

Extensive lysis of SB-HCC1,2,3 live tumor cell targets, but not of B16 melanoma targets, was observed when purified CD8 $^+$ IFN γ $^+$ T cells from mice treated with anti-PD-L1 ICI were co-cultured *in vitro* (compared to levels of lysis using CD8 $^+$ T cells from untreated SB-HCC- tumor bearing mice) (**Figure S7 A&B**). Consistent with a loss of both therapy (**Figure 7C**) and IFN γ anti-tumor CD8 $^+$ T cell response (**Figures S2&S6**) in mice treated with both anti-PD-L1 and VSV-IFN β , levels of SB-HCC1,2,3 cell lysis were not significantly different from those of untreated mice (**Figure S7 C**). In contrast, HCC-specific, CD8 $^+$ T cell-mediated lysis of HCC tumor targets was highly significant from mice treated with VSV-SB HCC1,2,3 alone or with VSV-SB HCC1,2,3 plus anti-PD-L1 (**Figures S7 D&E**). Finally, treatment with the melanoma derived VSV-cDNA ASMEL library plus anti-PD-L1 was not therapeutic against SB-HCC tumors (**Figure 7C**), did not induce an HCC-specific IFN γ $^+$ CD8 $^+$ T cell response (**Figure S6F**) and did not generate detectable CD8 $^+$ T cell lysis against SB-HCC tumor targets – although a potent CD8 $^+$ T cell lysis of murine melanoma cells was observed (**Figure S7 F**). Taken together, these data show that VSV-IFN β -SB-HCC1,2,3-mediated therapy of SB-HCC tumors correlated very well with generation of IFN γ $^+$ CD8 $^+$ T cells with strong lytic activity against HCC targets.

We apologize for not having made our selection process for IFN γ and IL-17 clearer in the text. As suggested by the Reviewer we performed a comprehensive cytokine measurement. We used a 26 cytokine panel configured from the Mouse Luminex® Discovery Assay (R&D Systems Catalog #: LXSAMSM) to compare levels of expression of different cytokines secreted from CD8 $^+$ T cells harvested from mice treated with anti-PDL-1 alone against SB-HCC1,2&3 live cell targets compared to CD8 $^+$ T cells from mice treated with VSV-IFN β -SB-HCC1,2,3. Of these 26 cytokines tested, we identified both IFN γ and IL17 as the only two cytokines whose levels were elevated significantly between anti-PD-LI treatment alone and VSV-IFN β -SB-HCC1,2,3 treatment. To clarify this in the revised manuscript we have added the following text to the **Results** on page 16, line 458:

In contrast, when splenocytes from mice treated with either VSV-SB-HCC1,2,3 alone (no anti-PD-L1 ICI), or with VSV-SB-HCC1,2,3 + anti-PD-L1 were re-stimulated *in vitro* with SB-HCC1,2,3 cells, IFN γ and IL-17 were identified as the only candidates from a screen of 26 different cytokines which were induced above that secreted by splenocytes from mice treated with ICI alone, indicating the induction of a Th17 recall response (**Figure 7D**).

6- *The authors need to study the liver microenvironment upon vaccination with the different groups. This should be done after the finish of the treatments (After day 30-32). The composition and phenotype of the cells needs to be shown by flow cytometry.*

The Reviewer's point is well taken. In this respect we have just started experiments in which we are using CyTOF, spatial transcriptomics and single cell RNAseq to study changes in the liver tumor microenvironment. These studies are actually an expansion on the Reviewer's suggestion and will study changes in the liver TME with time pre- (day 18-20), during (days 25-30) and post virus treatment both with, and without, anti-PD-L1 immune checkpoint blockade. These studies have just started and will take considerable time and resources to complete. Therefore, although we fully understand and agree with the Reviewer's point that such studies will be informative, we would request that we be allowed to save these studies for a future manuscript. This is because they will take a long time to complete, we feel we already have extensive amounts of data already in this revised manuscript and because these data are not necessarily central to the major thrust of the current manuscript which is showing that the provision of multiple HCC tumor antigens in VSV can generate a tumor curative CD8+ IFN γ + lytic T cell response.

Minor points:

1- *The method described for isolation of cells from liver tissue is incomplete and will not lead to high purity of cells. The authors need to clarify if this is the methodology used or if further purification of the cells is missing from the Methods.*

We apologize for the lack of clarity in our Methods section for liver cell isolation. We have now added these details to the **Methods** on page 28, line 806:

Mice were euthanized and livers were perfused with 20 mL PBS before being removed. Excised livers were dissected and passed through a 200-gauge stainless steel mesh. Cells were re-suspended in RPMI 1640/FCS. Cells were centrifuged (1500 r/min) and the resulting pellet resuspended in 40% Percoll solution with heparin. Cells were loaded on a 70% Percoll solution and centrifuged at 2000 r/min for 20 min at room temperature. Cells recovered from the gradient were centrifuged and washed with HBSS/FCS.

Response to Reviewer #2:

1. *There is generally a lack of detail in terms of the dosing and set-up of in vivo experiments. In the methods section there is no mention at all of use of virus in vivo, and*

the doses can often only be found in some of the figure legends. Doses of antibodies, particularly the immune cell-depleting antibodies are not provided at all. Most importantly, the critical information regarding the administration of virus is missing. It is mentioned once in the results for Figure 2 that the virus was injected IT. Was this the case for all experiments? If so, what exactly is the procedure? Are the mice subjected to laparoscopic procedure to inject into the tumors? If so, are the control mice also subjected to the same surgical intervention? Very importantly, is the virus injected into a single lesion or divided among multiple lesions? What volume is applied? Can any distinction be made in therapeutic effects in directly injected versus uninjected lesions?

We apologize for the lack of clarity in our methods. We have now added these details to the **Methods** on page 27, line 788:

Following hydrodynamic injection, animals were treated as described in the text. **Specific antibody doses were: anti-mouse PD-L1 (200 or 100µg per injection depending on the experiment, please see text) (BioX-Cell, clone 10F.9G2), CD8α (200µg/mouse) (BioX-Cell, clone 2.43), CD4 (200µg/mouse) (BioX-Cell, clone GK1.5), NK1.1 (200µg/mouse) (BioX-Cell, clone PK136), and IgG2A isotype (200µg/mouse) (BioX-Cell, clone 2A3). All antibody treatments were given i.v. by tail vein unless otherwise specified.**

In vivo treatment of SB-HCC tumor bearing mice with VSV-IFNβ was given by injection into the hepatic artery typically at 10⁷ pfu/dose delivered in 50µl volume (please see text for individual dosings). Therefore, virus was delivered equally across the tumor bearing livers rather than into specific lesions. Control mice were subjected to the same surgical interventions for injection of PBS or VSV-GFP. Other viruses (VSV-Lcn2, VSV-Lect2, or VSV-SMAGP) were given by the same route at either 10⁷ pfu/dose (VSV-Lcn2) or at 3x10⁶pfu/dose (combination of VSV-Lcn2, VSV-Lect2 and VSV-SMAGP).

2. It was mentioned that the maximum tolerated dose of VSV-IFNβ in patients was determined to be 3 x 10⁶ TCID₅₀, while the dose given in the described mouse studies was 10⁷ TCID₅₀. Can the authors comment on this large discrepancy in dosing, especially considering the massive difference in body weight of a mouse vs. human?

From data across a wide array of tumors and IND-enabling toxicology [Jenks N, *et al.*, Safety studies on intrahepatic or intratumoral injection of oncolytic vesicular stomatitis virus expressing interferon-beta in rodents and nonhuman primates. Human Gene Ther. 2010;21:451-62. PMC2865219; Willmon C, *et al.*, Vesicular stomatitis virus-induced immune suppressor cells generate antagonism between intra-tumoral oncolytic virus and cyclophosphamide. Mol Ther. 2011;19:140-9. PMC3017451] we conducted a first-in-human clinical study of VSV-hIFNβ (IT using ultrasound guidance into liver tumors) [NCT01628640]. Initially, 12 patients were treated. Treatment was mostly well tolerated with grade 1 pyrexia and myalgia being the most common adverse events (AE). A grade 5 (death) cytokine release syndrome (CRS) from sustained, supra-physiologic IFNβ transgene expression was observed in patient 12 at 3x10⁶ TCID₅₀ with large tumor burden (>80% liver involvement). Subsequently patients were restricted to <20% hepatic tumor involvement, with no further CRS. Since the re-start of the trial patients have been safely treated with doses up to 1.8x10⁷ median tissue culture infectious dose (TCID₅₀). With our

murine studies, we can administer VSV-GFP up to a dose of 5×10^7 without major systemic or neural toxicity. The addition of the IFN β gene as an added safety feature (please see response to point #3 below) allows this dose to be safely increased to above 10^9 which is why we used the VSV-IFN β virus in our first clinical trials (please see below)

3. Why was the vector encoding IFN β used for these experiments? Does IFN β exert any additional therapeutic effects in this model, or is this used simply for safety to reduce neurotoxic effects of VSV? Is it possible that the expression of IFN β is partially responsible for the loss of efficacy of anti-PD-L1 when administered in combination? Although differences in T cell responses of VSV-IFN β versus VSV-GFP are provided in Figure 4, it would be interesting to see how these vectors compare in terms of survival prolongation.

We used VSV-IFN β in these studies because this is the vector that we have already tested in the clinic and because the goal of the studies described in this work is to develop better combination therapies with immune checkpoint blockade building on the early trials with this virus. In our initial proposal to the FDA and DNA Recombinant Advisory Committee (RAC) to test VSV as an oncolytic, we hypothesized that VSV expressing IFN- β would be less cytolytic against normal cells and tissue (especially neural tissue) and therefore significantly attenuated *in vivo*, since these viruses would impede their own replication by activating the IFN antiviral pathway. However, since the IFN pathway is defective in transformed cells, VSV-IFN β should retain oncolytic activity by replicating efficiently. In addition, VSV-synthesized IFN could plausibly activate the IFN pathway in surrounding normal cells (enforcing protection against superfluous VSV), may exert antiproliferative effects against the tumor itself, and, finally, may enhance the antitumor immune response by stimulating NK and cytotoxic T cells as well as DC activity (as we published in Willmon C, *et al.*, Vesicular stomatitis virus-induced immune suppressor cells generate antagonism between intra-tumoral oncolytic virus and cyclophosphamide. *Mol Ther.* 2011;19:140-9. PMC3017451).

As part of the pre-clinical efficacy and toxicology studies performed to develop VSV-IFN β as a clinical agent (please see above), we compared VSV-GFP and VSV-IFN β in both immune competent and immune defective murine models and showed a significant enhancement of therapy in the immune competent setting of the VSV-IFN β virus over the VSV-GFP virus. As the Reviewer points out, **Figure 4** shows that VSV-GFP induces a very similar ‘squashing’ of the anti-tumor T cell response as does VSV-IFN β .

Therefore, to address the Reviewer’s point in the revised manuscript we have added the following text to the **Introduction** on page 5, line 94:

Therefore, here we sought to develop the VSV-IFN β virus based on our previous studies using it as a clinical agent in order to develop its use further in combination with standard of care and additional immunotherapeutic strategies.

4. Immune cells are characterized in several experiments in what is referred to as “liver tissue”. However, it is not explained whether this is a combination of tumor and liver tissue. Ideally, carefully isolated tumor tissue with minimal liver margins should be used for the analysis.

Immune cells were prepared from whole dissected tumor bearing livers and we did not dissect out tumor with margins from the mouse livers. Tumor explants were recovered from dissected tumor bearing livers by plating in culture and long term propagation *in vitro* (>3 weeks) during which time fibroblasts, hepatocytes and non-hepatocyte parenchymal cells die. Tumor identity of these lines (such as the three cell lines SB-HCC 1, 2 and 3) was validated by PCR for the hMet plasmid initially used to initiate tumor growth. As also requested by Reviewer #1 we have added the following text to the **Methods** on page 28, line 806 to clarify how the liver-derived immune cells were prepared:

Mice were euthanized and livers were perfused with 20 mL PBS before being removed. Excised livers were dissected and passed through a 200-gauge stainless steel mesh. Cells were re-suspended in RPMI 1640/FCS. Cells were centrifuged (1500 r/min) and the resulting pellet resuspended in 40% Percoll solution with heparin. Cells were loaded on a 70% Percoll solution and centrifuged at 2000 r/min for 20 min at room temperature. Cells recovered from the gradient were centrifuged and washed with HBSS/FCS.

5. How is the development of tumor confirmed prior to the start of treatment and how are mice stratified to treatment groups? It is presumed that tumor development is not 100% in this model and that there is a substantial amount of heterogeneity in terms of tumor progression. Ideally, the mice should be imaged prior to the start of treatment, and mice harboring tumors of a predefined size/number should be randomized to treatment groups such that average tumor burden is roughly equal among groups.

The Reviewer makes an excellent point and one which we have considered in detail. When initially setting up the Sleeping Beauty HCC model we spent much effort in trying to include an imaging component into the system. The closest we came was to express luciferase from one of the hMET or β -Catenin plasmids that are hydrodynamically injected into the livers. However, presumably due to the immunogenicity of the protein, we were never able to achieve more than 25-50% tumor take with luciferase expressing plasmids, whereas typically we achieve over 90% tumor take with just the SB/hMET/ β Catenin plasmids. Therefore, we were unable to use the system of tumor stratification and randomization as suggested by the Reviewer. Instead, we used histological examination in groups of mice from day 14, 21, 28, 35 to determine the extent and reproducibility of tumor take. These studies (part of which are shown in **Figure 1**) showed that tumor initiation is very reproducible between mice (15 of 16 mice tested had tumors by day 35) although the location and number of tumor foci differs considerably between mice. We observed small nests of transformed cells by histology in 4 of 4 mice at day 14; in a further 4 of 4 mice by day 21 clearly established foci of tumors were observed. Significantly, CD8⁺ T cell infiltration into livers was 10-fold higher in tumor-bearing mice than in non-tumor bearing control animals by day 22 indicating both presence of disease and an immune reaction to it (**Figure 1C,D**) – findings that we have substantiated by the addition of anti-SB-HCC CD8⁺ T cell response ELISpot data in our new **Supplemental Figures S1 and S2**. By day 28, 4 of 4 mice had multi focal disease and by day 35 3 of 4 mice had progressively expanding disease (one single mouse had no detectable tumors). So we agree completely with the Reviewer that the ideal situation would be to be able to image developing tumors and then randomize to treatment groups based on equal tumor size;

however, we were unable to establish such a screening system in this model and have used histological analysis to establish the reproducibility of tumor establishment as well as a time point for treatment (day 21 and later at which most mice reproducibly have signs of established multi-focal disease) even though there is clear variability between mice in the location and number of tumor foci.

6. It is stated that treatment was sometimes initiated on day 21, although the authors only show a first histological analysis of the liver on day 28, at which time tumors appear to be at a quite early stage of progression. Are tumors already evident on day 21? And is this a physiologically relevant time-point to initiate treatment considering the clinical presentation of the disease?

Please see our response to point #5 above with regards to the histological studies we undertook to determine when to start treatment. We believe that the situation at day 21 post plasmid injection, in which 100% of mice studied by day 21 had clearly established foci of tumors, represents a reasonable starting point for therapy as it relates to the human disease. With the success that we have shown now with the VSV-SB HCC 1,2&3 + anti-PD-L1 combination therapy to treat day 21 tumors (**Figure 7**), we now plan to extend the time period at which treatment starts (Day 30, 40 or 50) to see if the same regimen is still as effective at treating yet more established disease.

7. In the experimental set-up of the survival study shown in Figure 2d, anti-PD-L1 therapy is initiated on day 28, as opposed to day 21 in the previous experiments. It looks as if anti-PD-L1 therapy was no longer statistically significant in terms of survival prolongation in this treatment scheme (there are some long term survivors, but the non-survivors die at the same time as control mice). Therefore, the conclusion that the combination with VSV-IFN β abrogates the anti-PD-L1 treatment seems to be overreaching. Are differences in survival prolongation of anti-PD-L1 versus VSV-IFN β + anti-PD-L1 therapies statistically significant in this experiment?

In **Figure 2D** the day 21 therapy (see point #5 above for the rationale for this time point to initiate therapy) is virus (VSV-IFN β), whereas in **Figure 2A** the 21day timepoint therapy is anti-PD-L1 therapy (that is we kept the timepoint for therapy initiation the same). The Reviewer is absolutely correct that in **Figure 2D** there was no longer statistically significant survival prolongation for a subset of mice in the anti-PD-L1 treated group (that is the non-survivors die at the same time as control mice) but there was a consistent difference (over this and other multiple experiments) in the numbers of long-term survivors. We have now added data in response to Reviewer #1's points 1&5 above to show that this is because of the slowly developing anti-tumor T cell response which grows in magnitude until ~d28 and then starts to decline by ~day 49 in untreated mice, but which is maintained in strength through day 49 and then expanded in survivors through day 90 (new **Figures S1 & S2**). Therefore, the later that the immune checkpoint blockade therapy starts (day 21 to d28), the more likely it is that the weak anti-tumor T cell response will not be bolstered enough to impact survival. Therefore, to address the Reviewer's point we have now added new data in the new **Figure S1** on page 7, line 183:

To characterize the dynamics of the anti-tumor CD8+ T cell response in more detail, **Figure S1** shows that untreated mice bearing Sleeping Beauty HCC tumors developed CD8+ T cell responses against tumor explants which increased in magnitude with time (day 0 to day 28 following tumor initiation) using ELISPOT analysis. However, between day 28 and 49 these anti-SB-HCC CD8+ T cells responses waned significantly and by day 90 – at which time point most of the control mice have succumbed to the tumor – the anti-SB-HCC tumor CD8+ T cell response was low in the single surviving animal (**Figure S1.A**). When SB-HCC tumor-bearing mice were treated with anti-PD-L1 immune checkpoint blockade, anti SB-HCC CD8+ T cell responses were maintained in strength and even slightly increased through day 49 post tumor initiation. In addition, treatment with anti-PD-L1 ICB significantly enhanced the magnitude of the CD8+ T cell response by day 90 which was associated with the prolonged survival of mice to this time point (**Figure S1.B**).

To support this still further – that is addressing the bi-phasic nature of the survival curves - we have also added new data as explained in more detail in response to this Reviewer's point #11 below.

8. It is generally clear the oncolytic virus therapy has no therapeutic effect as a monotherapy in this tumor model. Can the authors comment on why that is the case? Could synergism of OV + ICI therapy be achieved if a more pronounced oncolytic effect could be produced?

Exactly as suggested by the Reviewer, we tested whether the lack of therapeutic effects seen with VSV-IFN β alone, or the combination of VSV-IFN β and anti-PD-L1, was the result of a lack of oncolysis by the virus. To investigate this, we have established 9 cell lines from tumor explants of SB-HCC tumor bearing mice of two different models (hMET+ β -Catenin driven-, or hMET+AKT driven-, tumors). All 9 of these explanted HCC cell lines support replication of VSV-IFN β at the same level, or in 3 cases significantly better, than do our control BHK cell line. In addition, even though the virus secretes its own (murine) IFN β , the further addition of universal Type I IFN to infected cultures was unable to inhibit viral replication from those lines. We have grown two of these explanted SB-HCC cell lines as subcutaneous tumors in C57Bl/6 mice and they are both insensitive to anti-PD-L1 therapy (in contrast to their behavior when growing in the liver in the SB-HCC model) and highly sensitive to VSV-IFN β treatment - but addition of anti-PD-L1 therapy does not enhance therapy in the subcutaneous model. These data suggest that the spontaneously arising SB-HCC tumors, which develop in the context of a fully intact murine immune system, have a highly defective tumor cell intrinsic response to Type I IFNs and support viral replication and oncolysis to a high level. Therefore, we believe that the liver/tumor-bearing liver microenvironment may exert heavily suppressive effects which are tumor cell extrinsic to prevent simple oncolysis from being effective in this model. In this respect we have just started experiments in which we are using CyTOF, spatial transcriptomics and single cell RNAseq to study changes in the liver tumor microenvironment. These studies will show changes in the liver TME with time pre- (day 18-20), during (days 25-30) and post virus treatment both with, and without, anti-PD-L1 immune checkpoint blockade.

To address the Reviewer's point here, we have added the following test to the **Discussion** on page 20, line 582:

We do not believe that the lack of efficacy of VSV-IFN β alone in this model is due to a lack of oncolytic capacity of the virus. This is because, of multiple cell lines established from tumor explants of SB-HCC tumor bearing mice, all support replication of, and oncolysis by, VSV-IFN β at high levels. Therefore, we hypothesize that, although spontaneously arising SB-HCC tumors which develop in the context of a fully intact murine immune system, have a highly defective response to Type I IFNs and the potential to support viral replication and oncolysis to a high level, the liver/tumor-bearing liver microenvironment may exert heavily suppressive, tumor cell-extrinsic effects to prevent simple oncolysis from being effective in this model.

9. In the experiments described in Figure 4, it seems that the experiment utilizing VSV-OVA employed a different dosing scheme than the experiments with VSV-GFP or VSV-IFN β . Therefore, it is difficult to compare the results. Similarly, the authors ask us to compare Figures 4G/H with Figures 4A/B (line 280), but it seems that the time-point for analysis was different (day 23 vs. 98). Therefore, this does not seem to be a fair comparison.

We thank the Reviewer for highlighting this. We performed the experiments with a shorter time frame in **Figures 4C&D** in order to accelerate our experiments (waiting for ~25 days instead of >50 days). We agree with the Reviewer that the direct comparison of the levels of anti-tumor (anti-OVA) T cells at days 23 (**Figure 4A/B**) with those of day 98 (**Figure 4G/H**) is not valid and we apologize for making that distinction in our original text. To clarify this in the revised manuscript we have amended our original text in the **Results** page 12, line 333:

When ICI was given in the absence of virus to allow for maximal expansion of the anti-tumor T cell response, treatment with VSV-OVA alone at a late stage (**Figure 4F**) generated ~~significantly greater high~~ numbers of anti-OVA T cells in both liver and spleen ~~compared to the levels spontaneously generated in response to growth of HCC-OVA tumors alone (Figures 4G&H). compared to Figures 4A&B~~. In addition, treatment with early anti-PD-L1 ICI combined with late VSV-OVA ~~significantly expanded the~~ generated anti-HCC_{TAA(OVA)} CD8⁺ T cells to high levels as a percentage of total CD8⁺ T cells in both liver and spleen (**Figures 4G&H**). Interestingly, the expansion of anti-HCC_{TAA(OVA)} CD8⁺ T cells by the addition of anti-PD-L1 ICI was significantly greater than the relative expansion of anti-VSV T cells (measured by tetramer directed against the immune-dominant VSV-N₅₂₋₅₉ peptide of the VSV N protein) (**Figures 4G&H**).

10. In Figure 4G/H, it is indicated that the tissue was harvested on day 98. However, if we look at the survival curves for this model, it seems that control mice begin dying at around 30 days post-induction, and most mice are already dead by day 98. Is analysis of tissue on day 98 not then inherently biased for the responding/surviving mice?

Again we agree completely with the Reviewer here. As part of our response to Reviewer 1 (point #1 above) we have added new data tracking the development of the (weak) anti-tumor response which is then maintained and expanded by anti-PD-L1 therapy. In addition, we have added data which shows that in those mice which are treated with anti-PD-L1, but which die early (at the same rate as the controls), the anti-PD-L1 ICI treatment has probably not achieved a therapeutic threshold of anti-tumour CD8+ T cell response (please see our response to point #11 below). Therefore, by sampling tissues at day 90 in **Figure 4G/H** we are inherently biasing the analysis for survival. To address this we have added the following text to the **Results** one page 12, line 340:

This condition of high anti tumor CD8+ T cell levels is almost certainly a biased analysis associated with survival of mice following ICI as opposed to that seen in mice which succumb to disease at early time points (see also **Figures S1&S2**).

11. In the survival curves depicted in Figures 6a and 7c, it is interesting that the mice diverge into two vastly different response phenotypes: either they die early (overlapping with control mice) or they have complete responses. Can the authors offer any explanation for this phenomenon? It is also difficult to see how these survival differences can be statistically significant compared to controls when the median 50% survival appears nearly identical.

In response to the Reviewer's point here, as well as in response to **Reviewer 1** point #1, we have added data tracking the anti-tumor CD8+ T cell response with time (new **Figures S1** and **S2**). In respect of the Reviewer's specific point here, we have also added new data as a new **Figure S5** one page 15, line 433:

In these experiments, as well as in those of **Figure 6A**, we observed a bi-phasic therapeutic response to anti-PD-L1 ICI in mice in which some anti-PD-L1 treated mice succumbed to disease early (up to day 50-60 post tumor initiation) at essentially the same rate as control treated mice, whilst a separate group of mice with ICI treatment were long term survivors (**Figures 6A&7C**). ELISPOT analysis showed that those ICI-treated mice which succumbed early to tumor all had low (<100 IFN γ spots per 10⁶ CD8+ T cells) anti-SB-HCC tumor CD8+ T cell responses, whilst long term survivors had significantly higher (>150 IFN γ spots per 10⁶ CD8+ T cells) anti-SB-HCC tumor CD8+ T cell responses (**Figure S5 A**). A similar analysis of the survival of individual untreated mice showed a trend in which the length of survival before succumbing to tumor was associated with the magnitude of the anti-SB-HCC CD8+ T cell response (**Figure S5 B**). Taken together, these data show that there exists a positive correlation between whether mice succumb to disease or are cured (>d150) and detectable levels of CD8+ anti-tumor T cell reactivity in mice either treated with anti-PD-L1 ICI or left untreated (**Figure S5 C**).

And we have added new **Discussion** on page 20, line 567:

Based on the data of **Figures S1, S2** and **S5**, we hypothesize that successful treatment of HCC with anti-PD-L1 treatment must maintain/reach a supra-threshold level of CD8+ anti-tumor T cell reactivity (>~150-200 IFN γ Spots/10⁶ CD8+ T cells). The reasons why

individual mice who are treated with apparently equivalent regimens of ICI either do, or do not, reach this therapeutic threshold of T cell reactivity are unclear; this may be related to purely technical issues associated with treatment administration. However, we believe that it is more likely to be related to the stochastic nature of tumor formation in this model in which hydrodynamic plasmid injections into the liver will be associated with different tumor phenotypes (genetically, antigenically and spatially). In turn, these data are significant in that they suggest the anti-tumor T cell response could be used as an important biomarker of likely patient response to ICI in the clinic.

12. In the experiments in which TAA-expressing vectors were used, it would have been helpful to characterize if/how the T cell responses were shifted back from anti-viral to anti-tumoral.

To address the Reviewer's point here, in combination with our response to **Reviewer 1, point #5**, we have added new data as a new **Figure S5** on page 18, line 497 of the **Results**:

CD8⁺ T cell dependent therapy with VSV-SB-HCC1,2,3 +/- ICI was highly associated with IFN γ -mediated anti-tumor effects. Thus, as before, although the anti-tumor T cell response to SB-HCC developed slowly and then waned after about 50 days, anti-PD-L1 ICB maintained the anti-tumor T cell response from d28-d49 and then significantly expanded it through day 90 (**Figures S5 A&B**). As in **Figure S2**, treatment with anti-PD-L1 and VSV-IFN β significantly diminished the strength of the anti-PD-L1-enhanced anti-tumor CD8⁺ T cell response and replaced it with a potent anti-VSV CD8⁺T cell response (**Figure S5 C**). Treatment with VSV-IFN β -SB-HCC1,2,3 alone (no ICB), which was itself highly therapeutic (**Figure 7C**), significantly enhanced the strength of the anti-tumor CD8⁺ T cell response early (d28 and 49) compared to that of the anti-PD-L1-enhanced CD8⁺ T cell response (**Figure S5 D**). Although the magnitude of this response did not fall off at a late timepoint (day 90), it was not as strong as that induced by anti-PD-L1 ICB alone at day 90 (**Figure S5 D**). In both cases (early and late) significant anti-tumor CD8⁺ IFN γ T cell responses co-existed with potent anti-viral (anti-VSV-N₅₂₋₅₉) T cell responses (**Figure S5 D**), which was not the case with treatment with VSV-IFN β (**Figure S5 C**). Treatment with anti-PD-L1+VSV-IFN β -SB-HCC1,2,3 induced a significantly enhanced and sustained (day 90) anti-tumor CD8⁺ T cell response compared to the anti-PD-L1-enhanced response and also co-existed with an anti-VSV CD8⁺ T cell response (**Figure S5 E**). These anti-SB-HCC CD8⁺ T cell responses did not cross react with murine melanoma cells indicating that the cDNA library approach generates a tumor specific immune response (**Figure S5 E**). This was confirmed by the observation that SB-HCC tumor bearing mice treated with anti-PD-L1 and the ASMEL VSV-cDNA library constructed from melanoma cDNA (as opposed to SB-HCC cDNA) (which was not therapeutic **Figure 7F**) significantly inhibited the magnitude of the anti-PD-L1 induced anti SB-HCC tumor CD8⁺ IFN γ + T cell response, although anti-B16 melanoma CD8⁺ IFN γ + T cells were abundant (**Figure S5 F**.)

13. Why is the low-dose anti-PD-L1 therapy seemingly effective in Figure 7f but not in 7c? In general, it is difficult to assess relative efficacy without the PBS or IgG controls.

The dose of anti-PD-L1 in **Figure 7F** was the high dose (200µg/mouse) and generated the typical effective ~50% survival that we observed reproducibly over multiple experiments (which we established using controls of PBS or IgG). In contrast, the dose in **Figure 7C** was low dose (100 µg/mouse) which we used to separate the survival curves and differentiate the added benefit of the VSV-IFNβ-SB-HCC1,2,3 library treatment in combination with anti-PD-L1 ICB. We have confirmed that these dose differences are stated in the Legends to **Figures 7C** and **7F**.

14. Is this tumor model prone to metastasis? If so, do the mice die from the primary tumors or from the metastases? This would be important to discriminate due to the OV therapy being administered locally to the hepatic tumors.

We have never observed systemic disease in SB-HCC bearing mice at the time of euthanasia due to liver tumor size and all the mice die of/are euthanized for local liver disease. We cannot say whether individual HCC tumors from one lesion in the liver spread to seed additional foci of disease within the liver. However, we have not observed tumors distant from the livers in any major organs (although we have not performed a detailed pathological analysis of all tissues).

Additional technical comments:

1. The anti-PD-L1 IHC images shown in Figure 1 are not clear and are difficult to draw conclusions from.

Thank you for drawing our attention to this concern. We have rebalanced the format of **Figure 1** to allow for significantly larger images for **Figure 1H**.

2. The images in 7A are rather small and difficult to see.

We have rebalanced the format of **Figure 7** to allow for significantly larger images for **Figure 7A**.

REVIEWERS' COMMENTS

Reviewer #1 (Remarks to the Author):

I have not further comments.

Reviewer #2 (Remarks to the Author):

The authors have added a substantial amount of new data and explanations to address the reviewers' concerns raised in the initial review. Although there still remain some open questions, the authors sufficiently acknowledge these, and despite these limitations, the work represents a large body of research that contributes important insights to the field. Overall, the quality of the manuscript has been substantially improved from the original version and, in this reviewer's opinion, could now be considered for publication in its current form.